# Peripheral Vision Transformer

**Juhong Min**[1]  **Yucheng Zhao**[2,3]  **Chong Luo**[2]  **Minsu Cho**[1]

[1]Pohang University of Science and Technology (POSTECH)
[2]Microsoft Research Asia (MSRA)
[3]University of Science and Technology of China (USTC)

http://cvlab.postech.ac.kr/research/PerViT/

## Abstract

Human vision possesses a special type of visual processing systems called *peripheral vision*. Partitioning the entire visual field into multiple contour regions based on the distance to the center of our gaze, the peripheral vision provides us the ability to perceive various visual features at different regions. In this work, we take a biologically inspired approach and explore to model peripheral vision in deep neural networks for visual recognition. We propose to incorporate peripheral position encoding to the multi-head self-attention layers to let the network learn to partition the visual field into diverse peripheral regions given training data. We evaluate the proposed network, dubbed PerViT, on ImageNet-1K and systematically investigate the inner workings of the model for machine perception, showing that the network learns to perceive visual data similarly to the way that human vision does. The performance improvements in image classification over the baselines across different model sizes demonstrate the efficacy of the proposed method.

## 1 Introduction

For the last ten years, convolution has been a dominant feature transformation in neural networks for visual recognition due to its superiority in modelling spatial configurations of images [21, 32, 34]. Despite the efficacy in learning visual patterns, the local and stationary nature of convolutional kernels limited the maximum extent of representation ability in flexible processing, *e.g.*, dynamic transformations with global receptive fields. Originally devised for natural language processing (NLP), self-attention [63] shed a light on this direction; equipped with adaptive input processing and the ability to capture long-range interactions, it has emerged as an alternative feature transform for computer vision, being widely adopted as a core building block [18]. The stand-alone self-attention models, *e.g.*, ViT [18], however, demand significantly more training data [57] for competitive performance with its convolutional counterparts [6, 25, 27, 72] since they miss certain desirable property which convolution possesses, *e.g.*, locality. These inherent pros and cons of convolution and self-attention encourage recent researches toward combinations of both so as to enjoy the best of the both worlds but which one suits the best for effective visual processing is yet controversial in literature [8, 9, 10, 12, 35, 37, 38, 40, 41, 49, 50, 59, 60, 62, 67, 69, 71, 73, 77].

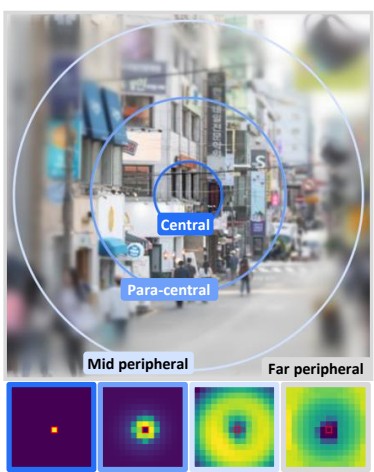

Figure 1: This work explores blending human peripheral vision (top) with attention-based neural networks (bottom) for visual recognition.

36th Conference on Neural Information Processing Systems (NeurIPS 2022).

Unlike the dominant visual feature transformations in machine vision, human vision possesses a special type of visual processing systems called *peripheral vision* [36]; it partitions the entire visual field into multiple contour regions based on the distances to the center of a gaze where each region identifies different visual aspects. As seen in Fig. 1, we have high-resolution processing near the center of our gaze, *i.e.*, central and para-central regions, to identify highly-detailed visual elements such as geometric shapes, and low-level details. For the regions more distant from the gaze, *i.e.*, mid and far peripheral regions, the resolution decreases to recognize abstract visual features such as motion, and high-level contexts. This systematic strategy enables us to effectively perceive important details within a small fraction (1%) of the visual field while minimizing unnecessary processing of background clutter in the rest (99%), thus facilitating efficient visual processing for human brain.

According to recent study on inner workings of vision transformers [12, 18, 49, 58, 71], their behaviors are related to the aforementioned visual processing strategies in the following respect; the attention maps of early layers are learned to locally capture *fine-grained geometric details* at *central regions* while those of later layers perform global attentions to identify *coarse-grained semantics and contexts* from the whole visual field, covering *peripheral regions*. This finding reveals that imitating biological designs may potentially help in modelling an effective machine vision, and also support recent approaches towards a hybrid [5, 10, 12, 24, 39] of convolution and self-attention beyond stand-alone visual processing to take advantages of the two different perception strategies: fine-grained/local and coarse-grained/global, similarly to the human visual processing as in Fig. 1.

In this work, we take a biologically inspired approach and propose to inject the peripheral inductive biases[1] to deep neural networks for image recognition. We propose to incorporate peripheral attention mechanism to the multi-head self-attention [63] to let the network learn to partition the visual field into diverse peripheral regions given training data where each region captures different visual features. We experimentally show that the proposed network models effective visual periphery for reliable visual recognition. Our main contributions can be summarized as follows:

- This work explores to narrow the gap between human and machine vision by injecting peripheral inductive biases to self-attention layers, and presents a new form of feature transformation named **M**ulti-head **P**eripheral **A**ttention (MPA).

- Based on the MPA, we introduce **Per**ipheral **Vi**sion **T**ransformer (PerViT), and systematically study the inner workings of PerViT by qualitatively and quantitatively analyzing its learned attentions, which reveal that the network learns to perceive visual elements similarly to the way that human vision does without any special supervisions.

- The performance improvements in image classification over columnar Transformer baselines, *e.g.*, DeiT, across different model sizes demonstrate the efficacy of the proposed approach.

## 2   Related Work

**Feature transformations in computer vision.** With notable success in NLP, Transformers [14, 63] introduced a paradigm shift in computer vision [3, 4, 7, 18, 30, 50, 55, 56, 58, 59, 62]. Despite their generalization capability, pure Vision Transformers [18] require extensive amount of training data to capture spatial layout of images due to lack of certain desirable property, *e.g.*, locality. This encouraged many recent ViT work to incorporate local inductive biases via distillation of convolutional biases [58], local self-attention [35, 40, 73], a hybridization [10, 12, 37], and augmenting convolutions [8, 60, 67, 69], all of which convey a unified message: "Despite high generalizibility of self-attentions, a sufficient amount of convolutional processing must be incorporated to capture the spatial configurations of images for reliable visual processing."

**Position encoding for Transformer.** Witnessing the efficacy of position encoding in capturing input structures in NLP [11, 28, 54], recent vision models [18, 50, 70] have begun employing position encodings for images to model spatial structures of images. In particular, relative position encoding (RPE) plays a vital role for the purpose: The work of Cordonnier *et al.* [9] proves that self-attention has close relationship with convolution when equipped with certain form of RPE. Wu *et al.* [70] explore the existing RPE methods used in vision transformers [11, 28, 54, 65] and draws a conclusion that RPEs impose convolutional processing on vision transformers. Dai *et al.* [10] observe that

---

[1]We refer peripheral inductive bias as the prioritization of our hypothesis which any attention-based neural networks can use to mimic human peripheral vision by modelling torus-shaped attentions as illustrated in Fig. 1.

depthwise convolution and self-attention can naturally be unified via RPEs. While offering promising directions, the previous RPE work, however, is limited in the sense that the focus of RPE utilization is restricted to only local attention, *e.g.*, convolution. This work exploits RPEs to devise an original visual feature transformation which naturally generalizes convolution and self-attention layers, thus enjoying the benefits of both via imitation of human visual processing system: *peripheral vision*.

**Peripheral vision for machine perception.** Along with central vision, peripheral vision plays a vital role in a wide range of visual recognition tasks [33]. The fundamental mechanisms of peripheral visual processing, however, have not been fully disclosed in human vision literature [52] which stimulated many researchers to reveal its inner workings and deep implications [2, 15, 16, 43, 53, 68]. The work of Rosenholtz [52] discusses pervasive myths and current findings about peripheral vision, suggesting that peripheral vision is more crucial for human perception than previously thought to perform diverse important tasks. Inspired by its importance, a number of pioneering work [16, 20, 22, 23, 44, 66] investigate the linkage between peripheral vision and machine vision, *e.g.*, CNNs, while some [31, 64] devise biologically-inspired models for the creation of stronger machine vision. Continuing previous study, this paper explores to blend human peripheral vision with attention-based neural networks, *e.g.*, vision transformer [18, 58], and introduces a new network called Peripheral Vision Transformer.

## 3 Our Approach

In this section, we introduce the Peripheral Vision Transformer (PerViT) which learns to model peripheral vision for effective image recognition. We first revisit the mathematical formulation of a self-attention layer and then describe how we improve it with peripheral inductive biases.

**Background: Multi-Head Self-Attention.** The multi-head self-attention (MHSA) [63] with $N_h$ heads performs an attention-based feature transformation by aggregating $N_h$ self-attention outputs:

$$\text{MHSA}(\mathbf{X}) := \underset{h \in [N_h]}{\text{concat}} \big[ \text{Self-Attention}^{(h)}(\mathbf{X}) \big] \mathbf{W}_{\text{out}} + \mathbf{b}_{\text{out}}, \tag{1}$$

where $\mathbf{X} \in \mathbb{R}^{HW \times D_{\text{emb}}}$ is a set of input tokens and $\mathbf{W}_{\text{out}} \in \mathbb{R}^{N_h D_h \times D_{\text{emb}}}$ and $\mathbf{b}_{\text{out}} \in \mathbb{R}^{D_{\text{emb}}}$ are the transformation parameters. The $N_h$ outputs of self-attention are designed to extract a diverse set of features from the input representation. Formally, the self-attention at head $h$ is defined as

$$\text{Self-Attention}^{(h)}(\mathbf{X}) := \text{Normalize} \big[ \Phi^{(h)}(\mathbf{X}) \big] \mathbf{V}^{(h)}, \tag{2}$$

where Normalize[·] denotes a row-wise normalization and $\Phi^{(h)}(\cdot) \in \mathbb{R}^{HW \times HW}$ is a function that provides spatial attentions based on content information to aggregate the values $\mathbf{V}^{(h)} = \mathbf{X}\mathbf{W}_{\text{val}}^{(h)}$:

$$\Phi^{(h)}(\mathbf{X}) := \exp(\tau \mathbf{X}\mathbf{W}_{\text{qry}}^{(h)}(\mathbf{X}\mathbf{W}_{\text{key}}^{(h)})^\top) = \exp(\tau \mathbf{Q}^{(h)}, \mathbf{K}^{(h)\top}), \tag{3}$$

using linear projections of $\mathbf{W}_{\text{qry}}^{(h)}, \mathbf{W}_{\text{key}}^{(h)}, \mathbf{W}_{\text{val}}^{(h)} \in \mathbb{R}^{D_{\text{emb}} \times D_h}$ for queries, keys, and values respectively where $\tau$ is softmax temperature and $\exp(\cdot)$ applies an element-wise exponential to the input matrix.

### 3.1 Multi-head Peripheral Attention

Based on the formulation of MHSA in Eq. 1, we define **M**ulti-head **P**eripheral **A**ttention (MPA) as

$$\text{MPA}(\mathbf{X}) := \underset{h \in [N_h]}{\text{concat}} \big[ \text{Peripheral-Attention}^{(h)}(\mathbf{X}, \mathbf{R}) \big] \mathbf{W}_{\text{out}} + \mathbf{b}_{\text{out}}, \tag{4}$$

where $\mathbf{R} \in \mathbb{R}^{HW \times HW \times D_r}$ is the relative position encoding with $D_r$ channel dimension. The self-attention in MHSA is now replaced with Peripheral-Attention(·), consisting of content- and position-based attention functions $\Phi_c^{(h)}(\mathbf{X}), \Phi_p^{(h)}(\mathbf{R}) \in \mathbb{R}^{HW \times HW}$, which is formulated as follows:

$$\text{Peripheral-Attention}^{(h)}(\mathbf{X}, \mathbf{R}) := \text{Normalize} \Big[ \Phi_c^{(h)}(\mathbf{X}) \odot \Phi_p^{(h)}(\mathbf{R}) \Big] \mathbf{V}^{(h)}, \tag{5}$$

where $\odot$ is Hadamard product which mixes the given pair of attentions to provide a mixed attention $\Phi_a^{(h)}(\mathbf{X}, \mathbf{R}) := \Phi_c^{(h)}(\mathbf{X}) \odot \Phi_p^{(h)}(\mathbf{R}) \in \mathbb{R}^{HW \times HW}$. For the content-based attention $\Phi_c$, we use exponentiated (scaled) dot-product between queries and keys as in Eq. 3: $\Phi_c^{(h)}(\mathbf{X}) := \exp(\tau \mathbf{Q}^{(h)}\mathbf{K}^{(h)\top})$. For the position-based attention $\Phi_p$, we design a neural network that aims to imitate human visual processing system, *e.g.*, peripheral vision, which we discuss next.

**Modelling peripheral vision: a Roadmap.** Human visual field can be grouped into several regions based on the Euclidean distances from the center of gaze, each forming ring-shaped region as seen in Fig. 1, where each region captures different visual aspects; the closer to the gaze, the more complex features we process, and further from the gaze, the simpler visual features we perceive. In the context of 2-dimensional attention map $\Phi_*(\cdot)_{\mathbf{q},:} \in \mathbb{R}^{HW}$, we refer the query position $\mathbf{q} \in \mathbb{R}^2$ as the center of gaze, *i.e.*, the position where feature of our interest lies at for the transformation. We refer the local regions around the query $\mathbf{q}$ as central/para-central regions and the rest as mid/far peripheral regions.

Perhaps the simplest approach to divide the visual field into multiple subregions is to perform a single linear projection on the Euclidean distances, *i.e.*, $\Phi_{\mathrm{p}}^{(h)}(\mathbf{R}) = \sigma\left[\mathbf{R}\mathbf{W}_{\mathrm{p}}^{(h)}\right]$ where $\mathbf{W}_{\mathrm{p}}^{(h)} \in \mathbb{R}^{D_{\mathrm{r}}}$ and $\sigma[\cdot]$ is non-linearity, similarly to the previous work of Wu *et al.* [70][2]. For straightforward imitation of peripheral vision, we use Euclidean distance for relative position input $\mathbf{R}$ and weigh the distances in $D_{\mathrm{r}}$ different ways for the network to learn the mapping in multiple scales: $\mathbf{R}_{\mathbf{q},\mathbf{k},:} :=$ concat$_{r \in [D_{\mathrm{r}}]}[w_{\mathrm{r}} \cdot \mathbf{R}_{\mathbf{q},\mathbf{k}}^{\mathrm{euc}}] \in \mathbb{R}^{D_{\mathrm{r}}}$ where $\{w_{\mathrm{r}}\}_{r \in [D_{\mathrm{r}}]}$ is a set of learnable parameters shared across layers and heads, and $\mathbf{R}_{\mathbf{q},\mathbf{k}}^{\mathrm{euc}} = \|\mathbf{q} - \mathbf{k}\|_2$ is the Euclidean distance between query and key positions $\mathbf{q}, \mathbf{k} \in \mathbb{R}^2$. In our experiments, we choose sigmoid function for $\sigma$ to provide normalized (positive) weights to the content-based attention $\Phi_{\mathrm{c}}$. A main drawback of this single-layer formulation is that $\Phi_{\mathrm{p}}$ is only able to provide Gaussian-like attention map as seen in top-left in Fig. 2, thus being unable to represent peripheral regions in diverse shapes. For the encoding function to represent various (torus-shaped) peripheral regions, the distances must be processed by an MLP:

$$\Phi_{\mathrm{p}}^{(h)}(\mathbf{R}) = \sigma\left[\mathrm{Linear}(\mathrm{ReLU}(\mathrm{Linear}(\mathbf{R};\mathbf{W}_{\mathrm{p1}}))\mathbf{W}_{\mathrm{p2}}^{(h)})\right] = \sigma\left[\mathrm{ReLU}(\mathbf{R}\mathbf{W}_{\mathrm{p1}})\mathbf{W}_{\mathrm{p2}}^{(h)}\right], \quad (6)$$

where $\mathbf{W}_{\mathrm{p1}} \in \mathbb{R}^{D_{\mathrm{r}} \times D_{\mathrm{hid}}}$ and $\mathbf{W}_{\mathrm{p2}}^{(h)} \in \mathbb{R}^{D_{\mathrm{hid}}}$ are the linear projection parameters[3], and ReLU gives non-linearity to the function. The first projection $\mathbf{W}_{\mathrm{p1}}$ is shared across the heads in order to exchange information so each function is able to provide attention that are effective or complementary to other heads' attention. Note that given identical relative distances between a fixed query point $\mathbf{q} \in \mathbb{R}^2$ and key points $\mathbf{k}_i, \mathbf{k}_j \in \mathbb{R}^2$, *i.e.*, $\mathbf{R}_{\mathbf{q},\mathbf{k}_i} = \mathbf{R}_{\mathbf{q},\mathbf{k}_j}$, Eq. 6 provides the same attention scores: $\Phi_{\mathrm{p}}(\mathbf{R})_{\mathbf{q},\mathbf{k}_i} = \Phi_{\mathrm{p}}(\mathbf{R})_{\mathbf{q},\mathbf{k}_j}$ as seen in top-right of Fig. 2. This property, however, is not always desired in practical scenarios because the rotational symmetric property hardly holds for most real-world objects. To break the symmetric property in Eq. 6 while preserving peripheral design to sufficient extent, we introduce *peripheral projections* in which the transformation parameters are given small spatial resolutions, *e.g.*, $K \times K$ window, such that $\mathbf{W}_{\mathrm{p1}} \in \mathbb{R}^{K^2 \times D_{\mathrm{r}} \times D_{\mathrm{hid}}}$ and $\mathbf{W}_{\mathrm{p2}}^{(h)} \in \mathbb{R}^{K^2 \times D_{\mathrm{hid}}}$ so that they provide similar but different attention scores, $\Phi_{\mathrm{p}}(\mathbf{R})_{\mathbf{q},\mathbf{k}_i} \neq \Phi_{\mathrm{p}}(\mathbf{R})_{\mathbf{q},\mathbf{k}_j}$, given $\mathbf{R}_{\mathbf{q},\mathbf{k}_i} = \mathbf{R}_{\mathbf{q},\mathbf{k}_j}$, by aggregating neighboring relative distances around the key location $\mathbf{k}$ as follows:

$$\Phi_{\mathrm{p}}^{(h)}(\mathbf{R})_{\mathbf{q},\mathbf{k},:} := \sigma\left[\sum_{\mathbf{n} \in \mathcal{N}(\mathbf{k})} \mathrm{ReLU}\left(\sum_{\mathbf{m} \in \mathcal{N}(\mathbf{k})} \mathbf{R}_{\mathbf{q},\mathbf{m},:}\mathbf{W}_{\mathrm{p1}\ \mathbf{m}-\mathbf{k},:,:}\right)\mathbf{W}_{\mathrm{p2}\ \mathbf{n}-\mathbf{k},:}^{(h)}\right], \quad (7)$$

where $\mathcal{N}(\mathbf{k}) := \left[\mathbf{k} - \lfloor\frac{K}{2}\rfloor, \ldots, \mathbf{k} + \lfloor\frac{K}{2}\rfloor\right] \times \left[\mathbf{k} - \lfloor\frac{K}{2}\rfloor, \ldots, \mathbf{k} + \lfloor\frac{K}{2}\rfloor\right]$ is a set of $K^2$ neighbors around input position $\mathbf{k}$. We set $K = 3$ for all layers and heads as $K > 3$ hardly brought improvements. Note that each linear projection in Eq. 7 is equivalent to a 4-dimensional convolution [51], taking 4-dimensional input $\mathbf{R} \in \mathbb{R}^{HW \times HW \times D_{\mathrm{r}}}$ to process in convolutional manner using 4-dimensional kernels in size of $K \times K \times 1 \times 1$, *i.e.*, $\mathbf{W}_{\mathrm{p1}} \in \mathbb{R}^{K \times K \times 1 \times 1 \times D_{\mathrm{r}} \times D_{\mathrm{hid}}}$. Precisely, the peripheral projection considers a small subset of 4D local neighbors that pivots the query position $\mathbf{q}$, similarly to the center-pivot 4D convolution [45, 46]. After each peripheral projection, we add an instance normalization layer [61] for stable optimization:

$$\mathbf{R}' = \mathrm{ReLU}\left(\mathrm{IN}(\mathrm{PP}(\mathbf{R};\mathbf{W}_{\mathrm{p1}});\boldsymbol{\gamma}_{\mathrm{p1}},\boldsymbol{\beta}_{\mathrm{p1}})\right), \quad \Phi_{\mathrm{p}}^{(h)}(\mathbf{R}) = \sigma\left(\mathrm{IN}(\mathrm{PP}(\mathbf{R}';\mathbf{W}_{\mathrm{p2}}^{(h)});\gamma_{\mathrm{p2}}^{(h)},\beta_{\mathrm{p2}}^{(h)})\right), \quad (8)$$

where $\boldsymbol{\gamma}_{\mathrm{p1}}, \boldsymbol{\beta}_{\mathrm{p1}} \in \mathbb{R}^{D_{\mathrm{hid}}}$ and $\gamma_{\mathrm{p2}}^{(h)}, \beta_{\mathrm{p2}}^{(h)} \in \mathbb{R}$ are weights/biases of the instance norms and $\mathrm{PP}(\cdot)$ denotes the peripheral projection: $\mathrm{PP}(\mathbf{R}, \mathbf{W})_{\mathbf{q},\mathbf{k},:} := \sum_{\mathbf{n} \in \mathcal{N}(\mathbf{k})} \mathbf{R}_{\mathbf{q},\mathbf{n},:}\mathbf{W}_{\mathbf{n}-\mathbf{k},:,:}$. The middle row of Fig. 2 depicts learned attentions of $\Phi_{\mathrm{p}}$ with peripheral projections, which provides peripheral attention maps in greater diversity compared to single- and multi-layer counterparts without $\mathcal{N}$.

---

[2]Given $\sigma[\cdot] := \exp(\cdot)$, Peripheral-Attention$^{(h)} = \mathrm{Normalize}[\exp(\mathbf{Q}^{(h)}\mathbf{K}^{(h)\top}) \odot \exp(\mathbf{R}\mathbf{W}_{\mathrm{p}}^{(h)})]\mathbf{V}^{(h)} = \mathrm{softmax}(\mathbf{Q}^{(h)}\mathbf{K}^{(h)\top} + \mathbf{R}\mathbf{W}_{\mathrm{rpe}})\mathbf{V}^{(h)}$, which is equivalent to the *bias mode* RPE presented in [70].

[3]We omit the bias terms in the linear layers for brevity.

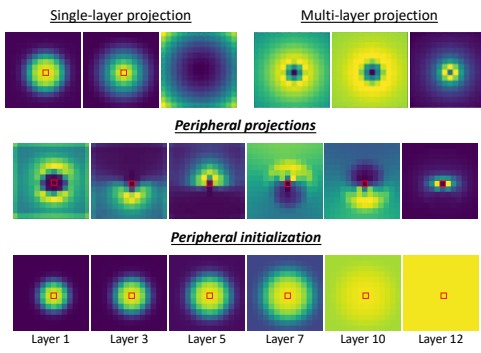

Single-layer projection   Multi-layer projection

*Peripheral projections*

*Peripheral initialization*

Layer 1  Layer 3  Layer 5  Layer 7  Layer 10  Layer 12

Figure 2: Top: representation ability of $\Phi_\text{p}$ under varying # layers. Middle: peripheral projections. Bottom: peripheral initialization.

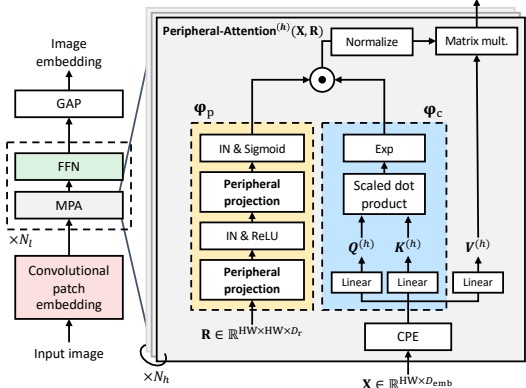

Figure 3: Overall architecture of PerViT which is based on DeiT [12] architecture.

**Peripheral initialization.** Recent study [49, 58] observe that early layers of trained vision transformer learn to attend locally whereas late layers perform global attentions. To facilitate training of our network, we inject this property in the beginning of the training stage by initializing parameters of $\Phi_\text{p}$ for the purpose, making attention scores near the queries larger than the distant ones in the early layers while uniformly distributing them in the late layers as seen in bottom row of Fig. 2. We refer this method as ***peripheral initialization*** for its resemblance to the functioning of peripheral vision [36] which also operates either locally or globally to perceive different visual features [76]. To formally put, given two arbitrarily chosen distances $\mathbf{R}^\text{euc}_{\mathbf{q},\mathbf{k}_i}, \mathbf{R}^\text{euc}_{\mathbf{q},\mathbf{k}_j} \in \mathbb{R}$ which satisfy $\mathbf{R}^\text{euc}_{\mathbf{q},\mathbf{k}_i} < \mathbf{R}^\text{euc}_{\mathbf{q},\mathbf{k}_j}$, we want $\Phi_\text{p}^{(l,h)}(\mathbf{R})_{\mathbf{q},\mathbf{k}_i} \gg \Phi_\text{p}^{(l,h)}(\mathbf{R})_{\mathbf{q},\mathbf{k}_j}$ [4] as $l \to 1$, *i.e.*, local attention in early layers, and $\Phi_\text{p}^{(l,h)}(\mathbf{R})_{\mathbf{q},\mathbf{k}_i} \approx \Phi_\text{p}^{(l,h)}(\mathbf{R})_{\mathbf{q},\mathbf{k}_j}$ as $l \to N_l$, *i.e.*, global attention in late layers, where $N_l$ is the total number of MPA layers. We first initialize the parameters of $\Phi_\text{p}^{(l,h)}$ and $\{w_r\}_{r\in[D_r]}$ to particular values. Specifically, for all layers $l \in [N_l]$ and heads $h \in [N_h]$,

$$w_r := -c_1, \forall r \in [D_\text{r}] \quad \mathbf{W}_\text{p1}^{(l)} := c_2 J_{K^2,D_r,D_\text{hid}} \quad \mathbf{W}_\text{p2}^{(l,h)} := c_2 J_{K^2,D_\text{hid}} \quad \boldsymbol{\gamma}_\text{p1}^{(l)} := \mathbf{1} \quad \boldsymbol{\beta}_\text{p1}^{(l)} := \mathbf{0} \tag{9}$$

where $c_1, c_2 \in \mathbb{R}^+$ are positive reals, and $J_{N,M} \in \mathbb{R}^{N \times M}$ refers to all-one matrix in size $N \times M$. The above initialization provides local attention after the second peripheral projection, *i.e.*, $\text{PP}(\mathbf{R}'; \mathbf{W}_\text{p2}^{(h)})_{\mathbf{q},\mathbf{k}_i,:} > \text{PP}(\mathbf{R}'; \mathbf{W}_\text{p2}^{(h)})_{\mathbf{q},\mathbf{k}_j,:}$ given $\mathbf{R}^\text{euc}_{\mathbf{q},\mathbf{k}_i} < \mathbf{R}^\text{euc}_{\mathbf{q},\mathbf{k}_j}$. Next, based on our findings that biases $\beta_\text{p2}^{(l,h)}$ and the weights $\gamma_\text{p2}^{(l,h)}$ in the second instance norm control the sizes and strengths of local attention respectively, we simulate peripheral initialization by setting their initial values as $\beta_\text{p2}^{(l,h)} := s_l$ and $\gamma_\text{p2}^{(l,h)} := v_l$ where respective $\{s_l\}_{l\in[N_l]}$ and $\{v_l\}_{l\in[N_l]}$ are sets of initial values for attention sizes and strengths. We set their values collected from uniform intervals: $s_l \in [-5.0, 4.0]$ and $v_l \in [3.0, 0.01]$ where $s_{l-1} < s_l$ and $v_{l-1} > v_l$ to give stronger local attentions to shallow layers compared to deep ones as seen in bottom row of Fig. 2. We set $c_1, c_2 = 0.02$ in our experiments. We refer the readers to the supplementary for the complete derivation of the peripheral initialization.

### 3.2 Peripheral Vision Transformer

Based on the proposed peripheral projections and initialization, we develop image classification models, dubbed Peripheral Vision Transformer, which is illustrated in Fig. 3. We follow similar architecture to DeiT [58] with convolutional patch embedding stem; as the original patchify stem [18] exhibits substandard optimizability due to its coarse-grained early visual processing [71], many recent ViT models adopt multi-resolution *pyramidal designs* [40, 67, 69, 73] to mitigate the issue. While the pyramidal models have shown their efficacy in learning reliable image embeddings, we stick with the original single-resolution *columnar design* for PerViT because features in multiple resolution make our study less interpretable, which further requires additional techniques for combining our

---

[4]We now use the superscript to indicate both layer and head indices for the ease of demonstration.

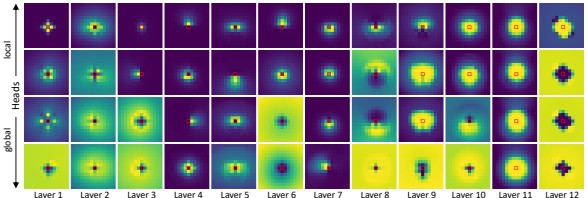

Figure 4: Learned attentions $\Phi_p$ of PerViT-T.

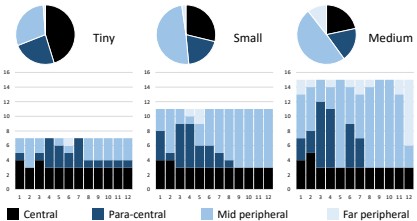

Figure 5: Peripheral region classification.

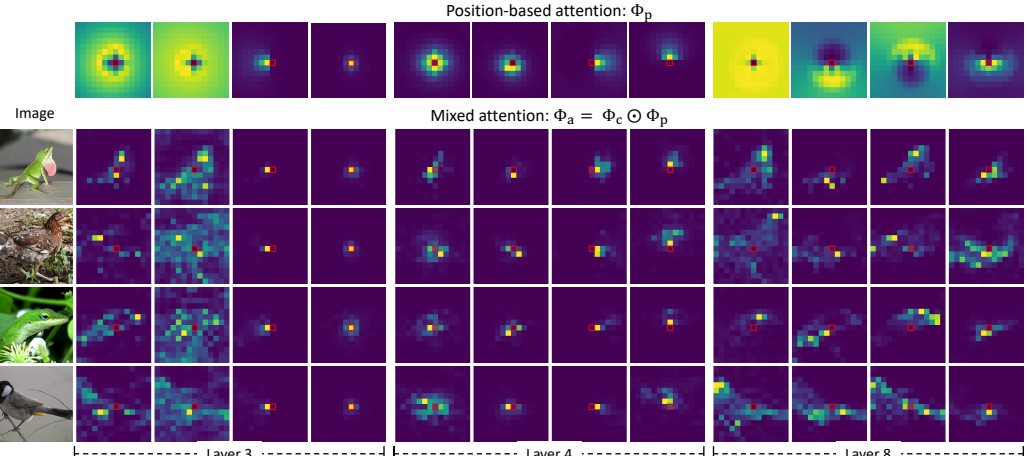

Figure 6: Visualization of learned attentions $\Phi_p$ and $\Phi_a$ for $l \in \{3, 4, 8\}$. Best viewed in electronics.

peripheral attention $\Phi_p$ with the existing cost-effective self-attentions mechanisms such as factorized attention [73], shifted-window [40], and cross-shaped window [17]. To carry out fine-grained early processing while keeping single-resolution features across the layers, we adopt convolutional patch embedding layer [71] with multi-stage layouts for channel dimensions. The convolutional embedding layer consists of four $3 \times 3$ and one $1 \times 1$ convolutions where the $3 \times 3$ convolutions are followed by batch norm [29] and ReLU [47]. For additional details, we refer to the supplementary materials.

**Overall pipeline.** Given an image, the convolutional patch embedding provides token embeddings $\mathbf{X}^{(1)} \in \mathbb{R}^{HW \times D_{emb}}$. Similarly to [18, 58], the embeddings are fed to a series of $N_l$ blocks each of which consists of an MPA layer and a feed-forward network with residual pathways:

$$\mathbf{X}^{(l')} = \text{MPA}(\text{LN}(\text{CPE}(\mathbf{X}^{(l)}))) + \mathbf{X}^{(l)}, \qquad \mathbf{X}^{(l+1)} = \text{FFN}(\text{LN}(\mathbf{X}^{(l')})) + \mathbf{X}^{(l')}, \qquad (10)$$

where LN is layer normalization [1], and FFN is an MLP consisting of two linear transformations with a GELU activation [26]. Following the work of [35, 73], we adopt convolutional position encoding (CPE), *i.e.*, a $3 \times 3$ depth-wise convolution, before first layer norm for its efficacy with negligible computational cost. The output $\mathbf{X}^{(N_h)}$ is global-average pooled to form an image embedding.

## 4 Experiments

In this section, we first investigate the inner workings of PerViT trained on ImageNet-1K classification dataset to examine how it benefits from the proposed peripheral projections and initialization, and then compare the method with previous state of the arts under comparable settings.

**Experimental setup.** Our experiments focus on image classification on ImageNet-1K [13]. Following training recipes of DeiT [58], we train our model on ImageNet-1K from scratch with batch size of 1024, learning rate of 0.001 using AdamW [42] optimizer, cosine learning rate decay scheduler, and the same data augmentations [14] for 300 epochs, including warm-up epochs. We evaluate our model with three different sizes, *e.g.*, Tiny (T), Small (S), and Medium (M). We use stochastic depths of 0.0, 0.1, and 0.2 for T, S, and M respectively. We refer to the supplementary for additional details.

### 4.1 The inner workings of PerViT

**Learning peripheral vision.** We begin by investigating how PerViT models peripheral vision by qualitatively analyzing its learned attention of $\Phi_\text{p}$. Figure 4 depicts the learned attention map of $\Phi_\text{p}^{(l,h)} \in \mathbb{R}^{HW}$ for all layers and heads where the query position is given at the center, *i.e.*, $\mathbf{q} = [7,7]^{\top}$[5]. We observe that the attentions are learned to be in diverse shapes of peripheral regions. Interestingly, without any special supervisions, the four attended regions ($N_h = 4$) in most layers are learned to complement each other to cover the entire visual field, capturing different visual aspects at each region (head), similarly to human peripheral vision illustrated in Fig. 1. For example, first two heads in Layer 3 attend the central regions while the others cover the rest peripheral regions. The second and third heads in Layer 8 cover top and bottom hemicircles respectively, forming a circular-shaped semi-global receptive field. Moreover, a large number of early attentions is in form of central/para-central regions while those of late layers are learned to cover mid to far peripheral regions. To quantitatively inspect how PerViT models the peripheral visual system, we classify every feature transformation layer in the network into one of the four visual regions $\mathbb{P} \in \{\text{c}, \text{p}, \text{m}, \text{f}\}$ where respective elements refer to central, para-central, mid, and far peripheral regions. PerViT-Attention of head $h$ at layer $l$ is classified as peripheral region $p$ if the average of its attention scores which fall in visual region $p$ is the largest among the others:

$$\text{PeripheralRegion}(l, h) := \underset{p \in \mathbb{P}}{\arg\max} \left[ \frac{1}{|\mathcal{P}|^2} \sum_{(\mathbf{q},\mathbf{k}) \in \mathcal{P} \times \mathcal{P}} \Phi_\text{p}^{(l,h)}{}_{\mathbf{q},\mathbf{k}} \cdot \mathbb{1} \left[ \|\mathbf{q} - \mathbf{k}\|_2 \in \mathcal{I}_p \right] \right], \quad (11)$$

where $\mathcal{P}$ is a set of spatial positions ($|\mathcal{P}| = HW$) and $\mathcal{I}_p$ is distance range (real-valued interval) of peripheral region $p$[6]. The pie charts of Fig. 5 describe the proportions of peripheral regions for Tiny, Small, and Medium models where the bar graphs show them in layer-wise manner[7]. Similarly to the visualized attention maps in Fig. 4, the early layers attends central/para-central regions whereas deeper ones focus on outer region. We observe that, as the model size grows, the number of mid/far peripheral attention increases whereas that of central/para-central attention stays similar, suggesting that the models no longer require local attentions once sufficient amount of processing is done in the central region because, we hypothesize, identifying geometric patterns, *e.g.*, corners and edges, is relatively simpler process than understanding high-level semantics.

**Inspecting the impact of attentions (static *vs.* dynamic).** To study how position-based attentions $\Phi_\text{p}$ contribute to the mixed attentions $\Phi_\text{a} = \Phi_\text{c} \odot \Phi_\text{p}$, we collect sample images and visualize their attention maps of Layers 3, 4 and 8 in Fig. 6. The mixed attentions $\Phi_\text{a}$ at Layer 4 are formed dynamically ($\Phi_\text{c}$) within statically-formed region ($\Phi_\text{p}$) while the attentions $\Phi_\text{a}$ at Layer 8 weakly exploit position information ($\Phi_\text{p}$) to form dynamic attentions ($\Phi_\text{c}$). The results reveal that $\Phi_\text{p}$ plays two different roles; it imposes ***semi-dynamic attention*** if the attended region is focused in a small area whereas it serves as ***position bias injection*** when the attended region is relatively broad. In the supplementary, we constructively prove that *an MPA layer in extreme case of semi-dynamic attention/position bias injection is in turn convolution/multi-head self-attention*, naturally generalizing the both transformations. To quantitatively examine the contributions of $\Phi_\text{c}$ and $\Phi_\text{p}$ to the mixed attention $\Phi_\text{a}$, we define a measure of 'impact' by taking inverse of difference between two attentions:

$$\Psi_\text{p}^{(l,h)} := [\|\Phi_\text{a}^{(l,h)} - \Phi_\text{p}^{(l,h)}\|_F]^{-1}, \quad (12)$$

where $\|\cdot\|_F$ is Frobenius norm. The higher the measure $\Psi_\text{p}^{(l,h)}$, the larger the impact of position-based attention $\Phi_\text{p}^{(l,h)}$. Being averaged over all test samples, $\Psi_\text{c}^{(l,h)}$ is similarly defined. As seen in Fig. 7, we observe a clear tendency that the impact of position-based attention is significantly higher in early processing, transforming features *semi-dynamically*, while the later layers require less position information, regarding $\Phi_\text{p}$ as a minor *position bias*. This tendency becomes more visible with larger models as seen in right of Fig 7; Small and Medium models exploit dynamic transformations much

---

[5]The columnar design of PerViT provides identical spatial resolution for every intermediate feature map in the network: $H, W = 14$, thus facilitating the ease of qualitative/quantitative analyses of the learned attentions.

[6]We use $\mathcal{I}_\text{c} = [0, 1.19)$, $\mathcal{I}_\text{p} = [1.19, 3.37)$, $\mathcal{I}_\text{m} = [3.37, 5.83)$, and $\mathcal{I}_\text{f} = [5.83, 7.9)$. We refer the readers to the supplementary materials for the justification on the these interval choices.

[7]We classify the $3 \times 3$ depth-wise convolution in CPE and the two linear projections in FFN as central regions as their receptive fields approximately fall in the interval of $\mathcal{I}_\text{c} = [0, 1.19)$.

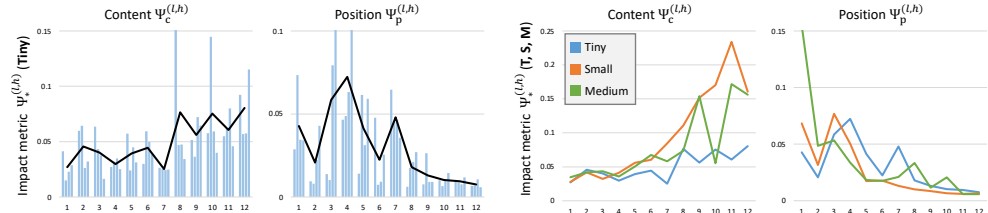

Figure 7: The measure of impact (x-axis: layer index, y-axis: the impact metric $\Psi_*$). Each bar graph shows the measure of a single head (4 heads at each layer), and the solid lines represent the trendlines which follow the average values of layers. (left: results of PerViT-T. right: results of T, S, and M.)

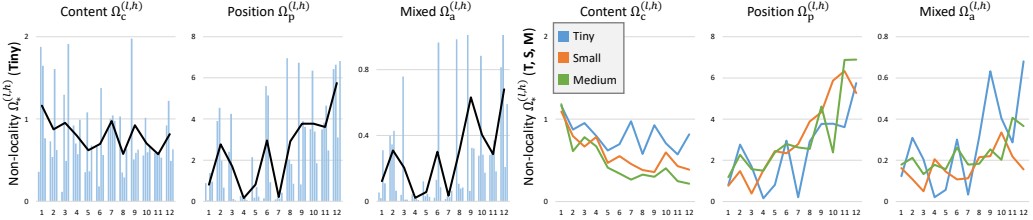

Figure 8: The measure of nonlocality (x-axis: layer index, y-axis: the nonlocality metric $\Omega_*$).

more compared to Tiny model, especially in the later layers. Moreover, we note that the impact measures of four heads (bar graphs) within each layer are unevenly distributed, showing high variance, which imply that the network evenly utilizes both position and content information simultaneously within each MPA layer as seen in Layer 3 in Fig. 6, *performing both static/local and dynamic/global transformation in a single-shot*. These results reveal that feature transformations for effective visual recognition should not be restricted to be in either position-only [25, 41] or content-only [18, 58] design but they should be in the form of a hybrid [10, 12].

**Inspecting the locality (local *vs.* global).** We further investigate the inner workings of PerViT by quantifying how locally $\Phi_*$ attends. Following the work of [12], we define the measure of nonlocality for $\Phi_p^{(l,h)}$ by summing all pair-wise query-key distances weighted by their attention scores:

$$\Omega_p^{(l,h)} := \frac{1}{|\mathcal{P}|^2} \sum_{(\mathbf{q},\mathbf{k}) \in \mathcal{P} \times \mathcal{P}} \Phi_{p\ \mathbf{q},\mathbf{k}}^{(l,h)} \mathbf{R}_{\mathbf{q},\mathbf{k}}^{\mathrm{euc}}. \tag{13}$$

The metrics $\Omega_c$ and $\Omega_a$ are similarly defined, being averaged over all test samples. As seen in Fig. 8, we observe a similar trend of locality between $\Phi_p$ and $\Phi_a$, which reveals the position information play more dominant role over the content information in forming spatial attentions ($\Phi_a$) for feature transformation. Interestingly, we also observe that content- and position-based attentions behave conversely; $\Phi_c$ attends globally in early layers, *i.e.*, large scores are distributed over the whole spatial region, while being relatively local in deeper layers. We hypothesize that the proposed $\Phi_p$ in early layers is trained to effectively suppress unnecessary scores of $\Phi_c$ at distant positions, thus exploiting only a few relevant ones within its local region of interest. Meanwhile, $\Phi_p$ at later layers gives $\Phi_c$ higher freedom in forming the spatial attention $\Phi_a$ as described in the plots of Fig. 7, which allows the attention scores of $\Phi_c$ to be clustered in semantically meaningful parts, *e.g.*, eyes of the animals as seen in attentions of the first head at Layer 8 (Fig. 6), which makes $\Phi_c$ relatively local.

## 4.2 Quantitative evaluation on ImageNet-1K

**Ablation study on main components.** In Tab. 1, we analyze the impact of each component in PerViT, which is denoted as (a), where C-stem refers to convolutional patch embedding stem[8]. We observe that the proposed attention $\Phi_p$ brings consistent gains to models (b, f, g, i) with relative improvements of 1.4~4.2%p for (a, d, e, h) respectively. Among three main components ($\Phi_p$, C-stem, CPE), $\Phi_p$ has the most significant impact on PerViT (a), losing 1.5%p Top-1 accuracy without $\Phi_p$, *i.e.*, model (b).

---

[8]We increase feature dimensions of the models (c) (without $\Phi_c$) and (f, h, i) (without C-stem) accordingly to make FLOPs comparable to the others (a-i) to ensure the accuracy drops are not simply due to lower FLOPs.

Table 1: Study on the effect of each component in PerViT.

| Reference | $\Phi_p$ | $\Phi_c$ | C-stem | CPE | Top-1 | Top-5 | FLOPs (G) |
|---|---|---|---|---|---|---|---|
| (a) | ✓ | ✓ | ✓ | ✓ | 78.8 | 94.3 | 1.6 |
| (b) | ✗ | ✓ | ✓ | ✓ | 77.3 | 94.1 | 1.6 |
| (c) | ✓ | ✗ | ✓ | ✓ | 76.8 | 93.5 | 1.6 |
| (d) | ✓ | ✓ | ✗ | ✓ | 77.8 | 94.0 | 1.5 |
| (e) | ✓ | ✓ | ✓ | ✗ | 78.1 | 94.0 | 1.6 |
| (f) | ✗ | ✓ | ✗ | ✓ | 76.3 | 93.2 | 1.5 |
| (g) | ✗ | ✓ | ✓ | ✗ | 76.7 | 93.3 | 1.6 |
| (h) | ✓ | ✓ | ✗ | ✗ | 76.5 | 93.4 | 1.5 |
| (i) | ✗ | ✓ | ✗ | ✗ | 72.3 | 93.4 | 1.5 |

Table 2: Comparisons between different relative position encodings with DeiT-Tiny [58] as a baseline.

| Method | Top-1 | FLOPS (G) |
|---|---|---|
| DeiT-T [58] | 72.2 | 1.3 |
| + CPVT [8] | 73.4 | 2.1 |
| + iRPE [70] | 73.7 | 1.1 |
| **+ PPE (ours)** | 74.4 | 1.1 |

Table 3: Ablation on PerViT-T/S/M: the effects of $\Phi_p$, C-Stem, and CPE.

| Ref. | $\Phi_p$ | C-Stem & CPE | T | S | M |
|---|---|---|---|---|---|
| (a) | ✓ | ✓ | 78.8 | 82.1 | 82.9 |
| (b) | ✗ | ✓ | 77.3 | 81.1 | 81.9 |
| (c) | ✗ | ✗ | 72.2 | 79.8 | 81.8 |

Table 4: Top-1 accuracy comparisons with DeiT-S [58] under different subsampling ratios: {100%, 50%, 25%}.

| Subsampling ratio | Top-1 | | Top-5 | |
|---|---|---|---|---|
| | DeiT-S | PerViT-S | DeiT-S | PerViT-S |
| 100% | 79.9 | 82.1 | 95.0 | 95.8 |
| 50% | 74.6 | 77.4 | 91.8 | 93.1 |
| 25% | 61.8 | 67.5 | 82.6 | 86.9 |

Surprisingly, PerViT without content-based attention $\Phi_c$, model (c), achieves decent performance, almost equalling to the accuracy of PerViT without position-based attention $\Phi_p$, model (b) (-0.5%p). The results verify that the proposed peripheral attention, which achieves comparable level of efficacy to the content-based attention, learns to generate reliable spatial attentions for visual recognition. We also implement the proposed position-based attention $\Phi_p$ on DeiT [58] baseline and compare the results with recent state-of-the-art RPE methods. As seen in Tab. 2, the large improvements over the previous RPE methods [8, 70] further verify the efficacy of the proposed peripheral position encoding (PPE). To confirm that the impact of $\Phi_p$ is consistent with large models, we conduct similar ablations using PerViT-S/M in Tab. 3; without $\Phi_p$, the accuracy consistently drops for all the three models. Comparing (b) with (c), we observe that C-Stem and CPE are less effective for large models, bringing 1.3%p and 0.1%p gains for Small and Medium respectively whereas they improve the Tiny model by 5.1%p. In contrast, the impact of $\Phi_p$ is consistent across different model sizes, bringing 1%p gains for all the three models. The better efficacy of $\Phi_p$ for larger models, we hypothesize, is due to its flexibility in modeling local/global spatial attentions while C-Stem/CPE are designed only to be local.

**Sample-efficiency of PerViT.** To investigate the training sample efficiency of our model, we train PerViT-S with ImageNet subsampled by fractions of 50% and 25%[9] and evaluate it on full-sized test set of ImageNet-1K. Table 4 compare our results with DeiT [58]; our model consistently surpasses the baseline for all subsampled datasets, showing its robustness under limited training data.

**Ablation study on $\Phi_p$.** The top section of Tab. 5 reports results of PerViT-T with different parameter initialization methods for $\Phi_p$ where peripheral denotes the proposed peripheral initialization, conv refers to convolutional initialization such that $s_l = -5.0$ and $v_l = 3.0$ for all $l \in [N_l]$, and rand refers to random initialization for all parameters in $\Phi_p$: $w_r$, $\mathbf{W}_{p1}^{(l)}$, $\mathbf{W}_{p2}^{(l,h)}$, $\gamma_{p1}^{(l)}$, $\beta_{p1}^{(l)}$, $\gamma_{p2}^{(l,h)}$, and $\beta_{p2}^{(l,h)}$. The results show the efficacy of our peripheral initialization which is also supported by the results in Fig. 4 and 8: $\Phi_p$ provides early local and late global attentions, suggesting that peripheral initialization effectively reduces burden in learning such form of attentions. The bottom section of Tab. 5 studies network designs for $\Phi_p$ where $\mathcal{N}$ represents the proposed peripheral projection, *i.e.*, projecting input distance representation by referring neighbors $\mathcal{N}$, ML refers to multi-layer design of $\Phi_p$, and Euc & Lrn indicate the type of embedding $\mathbf{R}$: Euc is relative Euclidean distances ($\mathbf{R}_{\mathbf{q},\mathbf{k},:} = \text{concat}_{r \in [D_r]}[w_r \cdot \mathbf{R}]$) where Lrn is relative distances between learnable vectors ($\mathbf{R} \in \mathbb{R}^{HW \times HW \times D_r}$). Without $\mathcal{N}$ and ML, we observe consistent accuracy drops for Euc and Lrn by 0.8%p and 0.2%p respectively. A sole multi-layer projection hardly improves accuracy but the model performs the best when $\mathcal{N}$ is jointly used, meaning that both need to complement each other to provide diverse attention shapes as in Fig 4. Furthermore, Euc models consistently surpasses Lrn models, implying the Euclidean distance is more straightforward encoding type than learnable vectors in capturing spatial configurations of images.

---

[9] For each subsamples, we increase the number of epochs to present models with a fixed number of images.

Table 5: Ablation study on different initialization methods (top section) and network designs (bottom section) for the position-based attention $\Phi_p$.

| Init. method for $\Phi_p$ | Top-1 | Top-5 |
|---|---|---|
| **Peripheral (ours)** | 78.8 | 94.3 |
| Conv | 78.6 | 93.8 |
| Rand | 78.5 | 93.6 |

| Network design for $\Phi_p$ | Top-1 | Top-5 |
|---|---|---|
| **Euc + $\mathcal{N}$ + ML (ours)** | 78.8 | 94.3 |
| Euc + ML | 77.9 | 94.0 |
| Euc | 78.0 | 94.0 |
| Lrn + $\mathcal{N}$ + ML | 77.8 | 94.0 |
| Lrn + ML | 77.5 | 93.8 |
| Lrn | 77.6 | 93.8 |

Table 6: Model performance on ImageNet-1K [13].

| | Model | Size (M) | FLOPs (G) | Top-1 (%) |
|---|---|---|---|---|
| Pyramidal Vision Transformers (*multi-resolution*) | PVT-T [67] | 13 | 1.9 | 75.1 |
| | CoaT-Lite-T [73] | 5.7 | 1.6 | 77.5 |
| | Swin-T [40] | 28 | 4.5 | 81.3 |
| | CoaT-Lite-S [73] | 20 | 4.0 | 81.9 |
| | Focal-T [74] | 29 | 4.9 | 82.2 |
| | Swin-S [40] | 50 | 8.7 | 83.0 |
| | CoaT-Lite-M [73] | 45 | 9.8 | 83.6 |
| | Focal-S [74] | 51 | 9.1 | 83.5 |
| Columnar Vision Transformers (*single-resolution*) | DeiT-T [58] | 5.7 | 1.3 | 72.2 |
| | XCiT-T12/16 [19] | 7.0 | 1.2 | 77.1 |
| | **PerViT-T (ours)** | 7.6 | 1.6 | 78.8 |
| | DeiT-S [58] | 22 | 4.6 | 79.8 |
| | T2T-ViT$_t$-14 [75] | 22 | 6.1 | 81.7 |
| | XCiT-S12/16 [19] | 26 | 4.8 | 82.0 |
| | **PerViT-S (ours)** | 21 | 4.4 | 82.1 |
| | DeiT-B [58] | 86 | 18 | 81.8 |
| | T2T-ViT$_t$-24 [75] | 64 | 15 | 82.6 |
| | XCiT-S24/16 [19] | 48 | 9.1 | 82.6 |
| | **PerViT-M (ours)** | 44 | 9.0 | 82.9 |

**Comparison with state of the arts.** Table 6 summarizes the results of our method and recent state of the arts. For fair comparison, the baselines used in our comparison are trained using $224 \times 224$ input resolution without distillations, and are grouped into either pyramidal or columnar ViT based on the network designs, *i.e.*, multi- or single-resolution feature processing, where the results are partitioned according to model sizes within each group. As shown in the bottom section of Tab 6, the proposed method achieves consistent improvements over the recent columnar ViT methods [12, 19, 58, 75] while showing competitive results to the pyramidal counterparts.

## 5 Scope and Limitations

Despite the interpretability and effectiveness of PerViT, it still leaves much room for improvements. First, PerViT-Attention (Eq. 5) is based on the original self-attention formulation [63], thus directly inheriting its limitations [18, 58], *e.g.*, quadratic complexity w.r.t. input resolution. The computational efficiency could be further improved by approximating low-rank matrices as in [7, 37, 73]. Second, given the ability to process high-resolution input with feasible complexity, the efficacy of PerViT could be improved by adopting multi-resolution pyramidal design following recent trend of ViT designs [17, 35, 37, 40, 67, 69, 73, 74]. Third, the focus of this paper is model development & exploration for image classification task but we believe the proposed idea is broadly generalizable to other vision applications such as object detection and segmentation. We leave this to future work.

## 6 Conclusion

This paper explores blending human peripheral vision with machine vision for effective visual recognition, and introduces Peripheral Vision Transformer which learns to provide diverse position-based attentions to model peripheral vision using peripheral projections and initialization. We have systematically investigated the inner workings of the proposed network and observed that the network enjoys the benefits of both convolution and self-attention by learning to decide level of the locality and dynamicity for the feature transformations, by the network itself given training data. The consistent improvements over the baseline models on ImageNet-1K classification across different model sizes and in-depth ablation study confirm the efficacy of the proposed approach.

## 7 Acknowledgments and Disclosure of Funding

This work was supported by the IITP grants (No.2021-0-01696: High-Potential Individuals Global Training Program (40%), No.2022-0-00290: Visual Intelligence for Space-Time Understanding and Generation based on Multi-layered Visual Common Sense (50%), No.2019-0-01906: AI Graduate School Program - POSTECH (10%)) funded by Ministry of Science and ICT, Korea. This work was done while Juhong Min was working as an intern at Microsoft Research Asia.

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
