# Peripheral Vision Transformers
# – *Supplementary Materials* –

**Juhong Min**[1]   **Yucheng Zhao**[2,3]   **Chong Luo**[2]   **Minsu Cho**[1]

[1]Pohang University of Science and Technology (POSTECH)
[2]Microsoft Research Asia (MSRA)
[3]University of Science and Technology of China (USTC)

http://cvlab.postech.ac.kr/research/PerViT/

In this supplementary material, we provide additional results and analyses of the proposed method, and implementation details. In Sec. A, we provide a complete derivation on the peripheral initialization introduced in Sec. 3.1 of our main paper. In Sec. B, we present additional details on peripheral region classification presented in Sec 4.1. In Sec. C, we prove that an Multi-head Peripheral Attention (MPA) in two extreme cases of semi-dynamic attention and position bias injection is in turn convolution and multi-head self-attnetion layers respectively. In Sec. D, we provide in-depth analyses of inner workings of PerViT and recent baselines of [18, 24] both quantitatively and qualitatively. In Sec. E and F, we provide network layouts of different model sizes, training hyperparameters, and implementation details. We conclude this paper with a short discussion on potential impacts of our work in Sec. G.

## A   Peripheral Initialization with a Complete Derivation

Recall the definition of the peripheral position encoding introduced in Sec. 3.1:

$$\mathbf{R}' \coloneqq \mathrm{ReLU}\left(\mathrm{IN}(\mathrm{PP}(\mathbf{R}; \mathbf{W}_{\mathrm{p1}}); \boldsymbol{\gamma}_{\mathrm{p1}}, \boldsymbol{\beta}_{\mathrm{p1}})\right), \qquad \Phi_{\mathrm{p}}^{(h)}(\mathbf{R}) \coloneqq \sigma\left(\mathrm{IN}(\mathrm{PP}(\mathbf{R}'; \mathbf{W}_{\mathrm{p2}}^{(h)}); \gamma_{\mathrm{p2}}^{(h)}, \beta_{\mathrm{p2}}^{(h)})\right), \tag{1}$$

where $\mathbf{R}_{\mathbf{q},\mathbf{k},:} \coloneqq \mathrm{concat}_{r\in[D_r]}[w_r \cdot \mathbf{R}_{\mathbf{q},\mathbf{k}}^{\mathrm{euc}}]$ is Euclidean distances between query $\mathbf{q}$ and key $\mathbf{k}$ position in $D_r$ different scales, $\boldsymbol{\gamma}_{\mathrm{p1}}^{(h)}, \boldsymbol{\beta}_{\mathrm{p1}}^{(h)} \in \mathbb{R}^{D_{\mathrm{hid}}}$, and $\gamma_{\mathrm{p2}}^{(h)}, \beta_{\mathrm{p2}}^{(h)} \in \mathbb{R}$ are weights/biases of the instance normalization layers [19] $\mathrm{IN}(\cdot)$, and $\mathbf{W}_{\mathrm{p1}} \in \mathbb{R}^{K^2 \times D_{\mathrm{r}} \times D_{\mathrm{hid}}}$ and $\mathbf{W}_{\mathrm{p2}}^{(h)} \in \mathbb{R}^{K^2 \times D_{\mathrm{hid}}}$ are learnable parameters of the peripheral projections $\mathrm{PP}(\cdot)$. Specifically, the peripheral projection with parameter $\mathbf{W} \in \mathbb{R}^{K^2 \times D_{\mathrm{in}} \times D_{\mathrm{out}}}$ transforms the input $\mathbf{R} \in \mathbb{R}^{HW \times HW \times D_{\mathrm{in}}}$ by referring neighboring distance representations in the key dimension as follows:

$$\mathrm{PP}(\mathbf{R}; \mathbf{W})_{\mathbf{q},\mathbf{k},:} \coloneqq \sum_{\mathbf{n}\in\mathcal{N}(\mathbf{k})} \mathbf{R}_{\mathbf{q},\mathbf{n},:}\mathbf{W}_{\mathbf{n}-\mathbf{k},:,:}, \tag{2}$$

where $\mathcal{N}(\mathbf{k})$ returns $K^2$ neighboring positions around position $\mathbf{k}$ including itself. Now assume

$$w_r \coloneqq -c_1, \qquad \mathbf{W}_{\mathrm{p1}} \coloneqq c_2 J_{K^2, D_{\mathrm{r}}, D_{\mathrm{hid}}}, \qquad \mathbf{W}_{\mathrm{p2}}^{(h)} \coloneqq c_2 J_{K^2, D_{\mathrm{hid}}}, \qquad \boldsymbol{\gamma}_{\mathrm{p1}} \coloneqq \mathbf{1}_{D_{\mathrm{hid}}}, \qquad \boldsymbol{\beta}_{\mathrm{p1}} \coloneqq \mathbf{0}_{D_{\mathrm{hid}}}, \tag{3}$$

for all $r \in [D_{\mathrm{r}}]$, and $h \in [N_h]$[1] where $c_1, c_2 \in \mathbb{R}^+$ are positive reals, $J_{N,M} \in \mathbb{R}^{N \times M}$ refers to all-one matrix in size $N \times M$, $\mathbf{1}_{D_{\mathrm{hid}}} = [1, ..., 1]^\top \in \mathbb{R}^{D_{\mathrm{hid}}}$, and $\mathbf{0}_{D_{\mathrm{hid}}} \in \mathbb{R}^{D_{\mathrm{hid}}}$ is a zero vector.

---

[1]The parameterization in Eq. 3 is applied for all the layers ($l \in [N_l]$) but we omit the layer indices for brevity.

36th Conference on Neural Information Processing Systems (NeurIPS 2022).

**Step 1.** Our first step is to prove the parameterization (Eq. 3) provides local attention after the second peripheral projection, *i.e.*, $\text{PP}(\mathbf{R}'; \mathbf{W}_{\text{p2}}^{(h)})_{\mathbf{q},\mathbf{k}_i,:} > \text{PP}(\mathbf{R}'; \mathbf{W}_{\text{p2}}^{(h)})_{\mathbf{q},\mathbf{k}_j,:}$ given $\mathbf{R}_{\mathbf{q},\mathbf{k}_i}^{\text{euc}} < \mathbf{R}_{\mathbf{q},\mathbf{k}_j}^{\text{euc}}$. Consider the first peripheral projection given $\mathbf{R} \in \mathbb{R}^{HW \times HW \times D_r}$:

$$\text{PP}(\mathbf{R}; \mathbf{W}_{\text{p1}})_{\mathbf{q},\mathbf{k},:} = \sum_{\mathbf{n} \in \mathcal{N}(\mathbf{k})} \mathbf{R}_{\mathbf{q},\mathbf{n},:} \mathbf{W}_{\text{p1}\ \mathbf{n}-\mathbf{k},:,:} \tag{4}$$

$$= c_2 \sum_{\mathbf{n} \in \mathcal{N}(\mathbf{k})} \mathbf{R}_{\mathbf{q},\mathbf{n},:} \cdot J_{D_r, D_{\text{hid}}} \tag{5}$$

$$= c_2 \sum_{\mathbf{n} \in \mathcal{N}(\mathbf{k})} \left( \sum_{r \in [D_r]} \mathbf{R}_{\mathbf{q},\mathbf{n},r} \right) \mathbf{1}_{D_{\text{hid}}} \tag{6}$$

$$= c_2 \sum_{\mathbf{n} \in \mathcal{N}(\mathbf{k})} \left( \sum_{r \in [D_r]} w_r \cdot \mathbf{R}_{\mathbf{q},\mathbf{n}}^{\text{euc}} \right) \mathbf{1}_{D_{\text{hid}}} \tag{7}$$

$$= -c_1 c_2 D_r \left( \sum_{\mathbf{n} \in \mathcal{N}(\mathbf{k})} \mathbf{R}_{\mathbf{q},\mathbf{n}}^{\text{euc}} \right) \mathbf{1}_{D_{\text{hid}}}, \tag{8}$$

which increases channel dimension to $D_{\text{hid}}$ and scales the Euclidean distance $\mathbf{R}_{\mathbf{q},\mathbf{k}}^{\text{euc}}$ by $-c_1 c_2 D_r$ after summing its neighbors. Note that the negation in Eq. 8 gives the inequality of

$$\text{PP}(\mathbf{R}; \mathbf{W}_{\text{p1}})_{\mathbf{q},\mathbf{k}_i,:} > \text{PP}(\mathbf{R}; \mathbf{W}_{\text{p1}})_{\mathbf{q},\mathbf{k}_j,:}, \tag{9}$$

given $\mathbf{R}_{\mathbf{q},\mathbf{k}_i}^{\text{euc}} < \mathbf{R}_{\mathbf{q},\mathbf{k}_j}^{\text{euc}}$ [2]. Given $\mathbf{R}^{\text{PP1}} := \text{PP}(\mathbf{R}; \mathbf{W}_{\text{p1}}) \in \mathbb{R}^{HW \times HW \times D_{\text{hid}}}$, the first instance norm $\text{IN}(\cdot; \mathbf{1}_{D_{\text{hid}}}, \mathbf{0}_{D_{\text{hid}}})$ provides normalized output as follows:

$$\text{IN}(\mathbf{R}^{\text{PP1}}; \mathbf{1}_{D_{\text{hid}}}, \mathbf{0}_{D_{\text{hid}}})_{\mathbf{q},\mathbf{k},c} := \frac{\mathbf{R}_{\mathbf{q},\mathbf{k},c}^{\text{PP1}} - \mathbb{E}_{\mathbf{m} \sim \mathcal{P}} \left[ \mathbf{R}_{\mathbf{q},\mathbf{m},c}^{\text{PP1}} \right]}{\mathbb{E}_{\mathbf{n} \sim \mathcal{P}} \left[ \left( \mathbf{R}_{\mathbf{q},\mathbf{n},c}^{\text{PP1}} - \mathbb{E}_{\mathbf{m} \sim \mathcal{P}} \left[ \mathbf{R}_{\mathbf{q},\mathbf{m},c}^{\text{PP1}} \right] \right)^2 \right]} \cdot \mathbf{1}_{D_{\text{hid}}\ c} + \mathbf{0}_{D_{\text{hid}}\ c}, \tag{10}$$

which simply normalizes each channel dimension to unit Gaussian while preserving the inequality, *i.e.*, $\text{IN}(\mathbf{R}^{\text{PP1}})_{\mathbf{q},\mathbf{k}_i,:} > \text{IN}(\mathbf{R}^{\text{PP1}})_{\mathbf{q},\mathbf{k}_j,:}$ for all $\mathbf{q}, \mathbf{k}_i, \mathbf{k}_j \in \mathcal{P}$. The ReLU nonlinearlity takes the normalized output and suppresses negative activations in $\mathbf{R}^{\text{IN1}} := \text{IN}(\mathbf{R}^{\text{PP1}}) \in \mathbb{R}^{HW \times HW \times D_{\text{hid}}}$, keeping the inequality only for positive activations:

$$\mathbf{R}'_{\mathbf{q},\mathbf{k}_i,:} = \text{ReLU}\left(\mathbf{R}^{\text{IN1}}\right)_{\mathbf{q},\mathbf{k}_i,:} > \text{ReLU}\left(\mathbf{R}^{\text{IN1}}\right)_{\mathbf{q},\mathbf{k}_j,:} = \mathbf{R}'_{\mathbf{q},\mathbf{k}_j,:}, \tag{11}$$

for $\mathbf{q}, \mathbf{k}_i, \mathbf{k}_j \in \mathcal{P}$ that satisfy $\mathbf{R}_{\mathbf{q},\mathbf{k}_i,:}^{\text{IN1}}, \mathbf{R}_{\mathbf{q},\mathbf{k}_j,:}^{\text{IN1}} > \mathbf{0}_{D_{\text{hid}}}$. Now consider second peripheral projection:

$$\mathbf{R}_{\mathbf{q},\mathbf{k}}^{\text{PP2}} = \text{PP}(\mathbf{R}'; \mathbf{W}_{\text{p2}}^{(h)})_{\mathbf{q},\mathbf{k}} \tag{12}$$

$$= \sum_{\mathbf{n} \in \mathcal{N}(\mathbf{k})} \mathbf{R}'_{\mathbf{q},\mathbf{n},:} \mathbf{W}_{\text{p2}\ \mathbf{n}-\mathbf{k},:}^{(h)} \tag{13}$$

$$= c_2 \sum_{\mathbf{n} \in \mathcal{N}(\mathbf{k})} \mathbf{R}'_{\mathbf{q},\mathbf{n},:} \cdot \mathbf{1}_{D_{\text{hid}}} \tag{14}$$

$$= c_2 \sum_{\mathbf{n} \in \mathcal{N}(\mathbf{k})} \left( \sum_{h \in [D_{\text{hid}}]} \mathbf{R}'_{\mathbf{q},\mathbf{n},h} \right) \tag{15}$$

$$= c_2 D_{\text{hid}} \sum_{\mathbf{n} \in \mathcal{N}(\mathbf{k})} \mathbf{R}'_{\mathbf{q},\mathbf{n},d} \tag{16}$$

for any $d \in [D_{\text{hid}}]$. Similarly to the first $\text{PP}(\cdot)$, Eq. 16 scales the input by $c_2 D_{\text{hid}} \in \mathbb{R}^+$ after summing its neighbors. Hence, $\mathbf{R}_{\mathbf{q},\mathbf{k}_i}^{\text{PP2}} > \mathbf{R}_{\mathbf{q},\mathbf{k}_j}^{\text{PP2}}$ holds given $\mathbf{R}_{\mathbf{q},\mathbf{k}_i}^{\text{euc}} < \mathbf{R}_{\mathbf{q},\mathbf{k}_j}^{\text{euc}}$ if $\mathbf{R}_{\mathbf{q},\mathbf{k}_i}^{\text{PP2}}, \mathbf{R}_{\mathbf{q},\mathbf{k}_j}^{\text{PP2}} > 0$, providing local attention. ∎

---

[2] Note that $\sum_{\mathbf{n} \in \mathcal{N}(\mathbf{k}_i)} \mathbf{R}_{\mathbf{q},\mathbf{n}}^{\text{euc}} < \sum_{\mathbf{n} \in \mathcal{N}(\mathbf{k}_j)} \mathbf{R}_{\mathbf{q},\mathbf{n}}^{\text{euc}}$ as we deal with Euclidean distances.

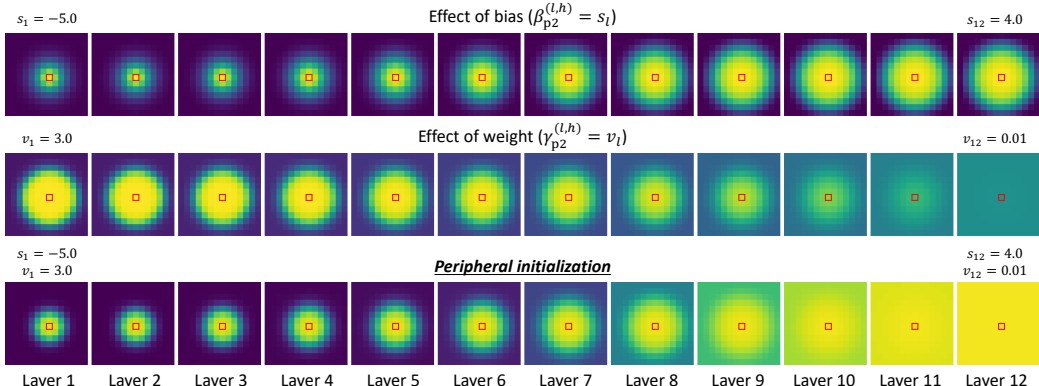

Figure S1: Effects of the biases $\beta_{\text{p2}}^{(l,h)}$ (top) and weights $\gamma_{\text{p2}}^{(l,h)}$ (middle) to the size and strength of local attentions. Our network exploits early local and late global attentions in the beginning of the training stage, which are achieved via peripheral initialization (bottom).

**Step 2.** We now show that the respective size and strength of local attention in the peripheral position encoding $\Phi_{\text{p}}^{(h)}$ are controlled by the bias $\beta_{\text{p2}}^{(h)}$ and weight $\gamma_{\text{p2}}^{(h)}$ of the second instance norm, which is defined as follows:

$$\text{IN}(\mathbf{R}^{\text{PP2}}; \gamma_{\text{p2}}^{(h)}, \beta_{\text{p2}}^{(h)})_{\mathbf{q},\mathbf{k}} := \frac{\mathbf{R}_{\mathbf{q},\mathbf{k}}^{\text{PP2}} - \mathbb{E}_{\mathbf{m}\sim\mathcal{P}}\left[\mathbf{R}_{\mathbf{q},\mathbf{m}}^{\text{PP2}}\right]}{\mathbb{E}_{\mathbf{n}\sim\mathcal{P}}\left[\left(\mathbf{R}_{\mathbf{q},\mathbf{n}}^{\text{PP2}} - \mathbb{E}_{\mathbf{m}\sim\mathcal{P}}\left[\mathbf{R}_{\mathbf{q},\mathbf{m}}^{\text{PP2}}\right]\right)^2\right]} \cdot \gamma_{\text{p2}}^{(h)} + \beta_{\text{p2}}^{(h)} \tag{17}$$

$$= \mathbf{R}_{\mathbf{q},\mathbf{k}}^{\text{PP2-Norm}} \cdot \gamma_{\text{p2}}^{(h)} + \beta_{\text{p2}}^{(h)}, \tag{18}$$

where $\mathbf{R}^{\text{PP2-Norm}}$ refers to unit-Gaussian normalized $\mathbf{R}^{\text{PP2}}$. Note that the weight and bias terms in the above formulation (Eq. 18) have immediate control over the distribution of the position-based attention scores:

$$\Phi_{\text{p}\ \mathbf{q},\mathbf{k}}^{(h)} = \sigma\left(\mathbf{R}_{\mathbf{q},\mathbf{k}}^{\text{PP2-Norm}} \cdot \gamma_{\text{p2}}^{(h)} + \beta_{\text{p2}}^{(h)}\right). \tag{19}$$

For example, given some fixed query position $\mathbf{q}$, a large bias term encourages global attention:

$$\lim_{\beta_{\text{p2}}^{(h)}\to\infty} \Phi_{\text{p}\ \mathbf{q},\mathbf{k}}^{(h)} = \lim_{\beta_{\text{p2}}^{(h)}\to\infty} \sigma\left(\mathbf{R}_{\mathbf{q},\mathbf{k}}^{\text{PP2-Norm}} \cdot \gamma_{\text{p2}}^{(h)} + \beta_{\text{p2}}^{(h)}\right) = 1 \quad \text{for all} \quad \mathbf{k} \in \mathcal{P}, \tag{20}$$

controlling the size of local attention as seen in the top row of Fig. S1, whereas small magnitude of the weight term makes attention distribution more uniform:

$$\lim_{\gamma_{\text{p2}}^{(h)}\to 0} \Phi_{\text{p}\ \mathbf{q},\mathbf{k}}^{(h)} = \lim_{\gamma_{\text{p2}}^{(h)}\to 0} \sigma\left(\mathbf{R}_{\mathbf{q},\mathbf{k}}^{\text{PP2-Norm}} \cdot \gamma_{\text{p2}}^{(h)} + \beta_{\text{p2}}^{(h)}\right) = \sigma\left(\beta_{\text{p2}}^{(h)}\right) \quad \text{for all} \quad \mathbf{k} \in \mathcal{P}, \tag{21}$$

manipulating the strength of the local attention as seen in the middle row of Fig. S1. $\blacksquare$

**Simulating peripheral initialization.** Figure S1 illustrates attention maps $\Phi_{\text{p}\ \mathbf{q},:}^{(l,h)} \in \mathbb{R}^{HW}$ given a query position at the center, *i.e.*, $\mathbf{q} = [7,7]^\top$, under varying biases $\beta_{\text{p2}}^{(l,h)} = s_l$ and weights $\gamma_{\text{p2}}^{(l,h)} = v_l$ where the respective values are collected from uniform intervals: $s_l \in [-5.0, 4.0]$ and $v_l \in [3.0, 0.01]$ which satisfy $s_{l-1} < s_l$ and $v_{l-1} > v_l$. The proposed peripheral initialization imposes strong locality in early layers ($s_1 = -5.0$ and $v_1 = 3.0$) and global attention ($s_{12} = 4.0$ and $v_{12} = 0.01$) in late layers, facilitating training of PerViT as demonstrated in Sec. 4.1 with Figs. 4 and 8 of our main paper.

# B  Peripheral Region Classification

In Sec. 4.1 of the main paper, we have classified every feature transformation layer in PerViT into one of the four visual regions $\mathbb{P} \in \{c, p, m, f\}$ where the elements refer to central, para-central, mid, and far peripheral regions respectively. We use the following formulation:

$$\text{PeripheralRegion}(l, h) := \underset{p \in \mathbb{P}}{\arg\max} \left[ \frac{1}{|\mathcal{P}|^2} \sum_{(\mathbf{q}, \mathbf{k}) \in \mathcal{P} \times \mathcal{P}} \Phi_p^{(l,h)}{}_{\mathbf{q}, \mathbf{k}} \cdot \mathbb{1} \left[ \|\mathbf{q} - \mathbf{k}\|_2 \in \mathcal{I}_p \right] \right], \quad (22)$$

where we set $\mathcal{I}_c = [0, r_c)$, $\mathcal{I}_p = [r_c, r_p)$, $\mathcal{I}_m = [r_p, r_m)$, and $\mathcal{I}_f = [r_m, r_f)$ in experiments.

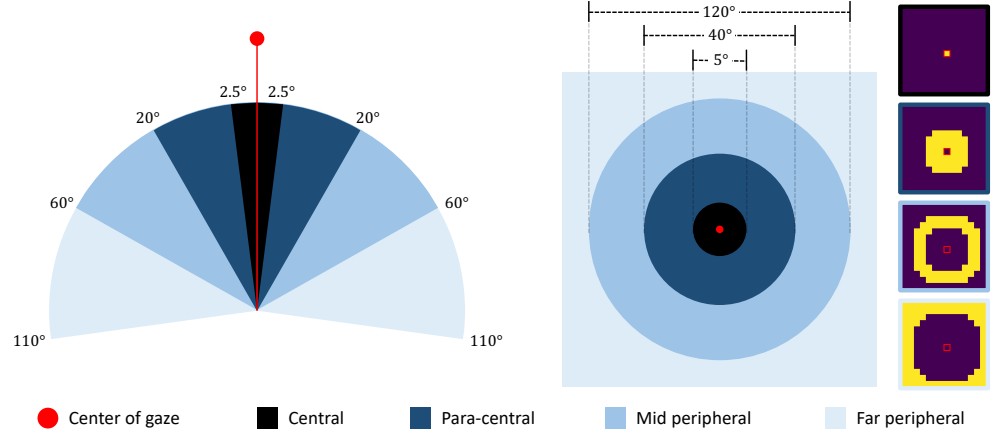

Figure S2: Peripheral vision of human eye (left). Peripheral vision of an MPA layer in PerViT (right).

According to vision science literature [15], *e.g.*, physiology, ophthalmology, and optometry, the maximum extent of whole visual field is bounded within 100-110° (horizontal) angles from the gaze, providing 200-220° perceivable area like fan-shaped figure in the left of Fig. S2. The central vision, which we call central *region* in the context of attention map, refers to area corresponding 5° of whole visual field whereas the paracentral vision covers upto 8°. The near-, mid-, and far-peripheral vision correspond to non-overlapping areas outside circles 8°, 60°, and 120° in diameter respectively. When classifying the feature transformation layers in Sec. 4.1, we adhere to these angular ratios to imitate human peripheral vision as faithfully as possible:

$$\theta_p : 220° = \pi r_p^2 : HW, \quad (23)$$

where $\theta_p \in \{5°, 40°, 120°, 220°\}$ is a set of angles that divide visual field into four non-overlapping areas of central, para-central[3], mid-, and far-peripheral regions respectively. Solving the equality in Eq. 23 with respect to the radius $r_p$ gives

$$r_p = \sqrt{\frac{HW \cdot \theta_p}{220° \pi}}. \quad (24)$$

We then have $r_c = 1.19$, $r_p = 3.37$, $r_m = 5.83$, and $r_f = 7.9$, *i.e.*, $\mathcal{I}_c = [0, 1.19)$, $\mathcal{I}_p = [1.19, 3.37)$, $\mathcal{I}_m = [3.37, 5.83)$, and $\mathcal{I}_f = [5.83, 7.9)$. For the classification experiments performed in Fig. 5 of the main paper, we classify two linear projections in an MLP and a $3 \times 3$ convolution in CPE as central regions as radii of their receptive fields approximately fall in the interval of $\mathcal{I}_c = [0, 1.19)$.

We consider each query location $\mathbf{q}$ of MPA, *e.g.*, the position of a feature we want to transform, as a focal point, assuming each MPA in PerViT simultaneously processes $H \times W$ pixel locations with $H \times W$ different focal points given input feature size of $H \times W$. This assumption provides ring-shaped attentions if a query is located at the center of the feature map (Fig. S2). While we have developed our narrative in the context of images (2D), the assumption deviates from the reality when considering FOV of a physical eyeball (3D).

---

[3]We found that Eq. 23 gives radius of 1.5 given para-central angle of 8°, resulting in quite narrow interval: $[1.19, 1.5)$. Since the concept of paracentral (8°) and near-peripheral (60°) visions are used interchangeably in literature [22], we set the angle for para-central region to some intermediary value between 8° and 60°, *i.e.*, $\theta_p = 40°$, so every peripheral region gets radii of (approximately) equal size as seen in the right of Fig. S2.

## C  Proof of MPA as Convolution and MHSA

In this section, we constructively prove that a Multi-head Peripheral Attention (MPA) layer in extreme case of semi-dynamic attention, *i.e.*, strong attention at a small area, and position bias injection, *i.e.*, relatively broad attention over the whole visual field, is in turn convolution and multi-head self-attention layers respectively. We first recall the definition of the MPA:

$$\text{MPA}(\mathbf{X}) \coloneqq \underset{h \in [N_h]}{\text{concat}} \big[\text{Peripheral-Attention}^{(h)}(\mathbf{X}, \mathbf{R})\big] \mathbf{W}_{\text{out}} + \mathbf{b}_{\text{out}} \tag{25}$$

$$= \left( \sum_{h \in [N_h]} \text{Peripheral-Attention}^{(h)}(\mathbf{X}, \mathbf{R}) \right) \mathbf{W}_{\text{out}}^{(h)} + \mathbf{b}_{\text{out}}, \tag{26}$$

where $\mathbf{W}_{\text{out}}^{(h)} = (\mathbf{W}_{\text{out}})_{(h-1)D_h+1:hD_h+1} \in \mathbb{R}^{D_h \times D_{\text{emb}}}$ and Peripheral-Attention is defined as

$$\text{Peripheral-Attention}^{(h)}(\mathbf{X}, \mathbf{R}) \coloneqq \text{Normalize}\left[ \Phi_{\text{c}}^{(h)}(\mathbf{X}) \odot \Phi_{\text{p}}^{(h)}(\mathbf{R}) \right] \mathbf{V}^{(h)}. \tag{27}$$

**MPA as a convolution.** Assume the position-based function at each head is learned to perform 'hard attention' on one of its surrounding positions, *i.e.*, *an extreme semi-dynamic attention*. To formally put, $\Phi_{\text{p}}^{(h)}(\mathbf{R})_{\mathbf{q},\mathbf{k}} \coloneqq \mathbb{1}[0 = s(h) - \mathbf{R}_{\mathbf{q},\mathbf{k},:}] \in \mathbb{R}^{HW \times HW}$ where $\mathbf{R} \in \mathbb{R}^{HW \times HW \times 2}$ is a matrix containing pair-wise offsets between query and key positions, *i.e.*, $\mathbf{R}_{\mathbf{q},\mathbf{k}} \coloneqq \mathbf{q} - \mathbf{k}$, and $s(h) :$ $[N_h] \to \nabla_k$ is a bijective mapping of heads onto a fixed set of offsets $\nabla_k = \{-\lfloor k/2 \rfloor, ..., \lfloor k/2 \rfloor\} \times \{-\lfloor k/2 \rfloor, ..., \lfloor k/2 \rfloor\}$, *i.e.*, $N_h = k^2$. Given the assumptions, consider Peripheral-Attention at query position $\mathbf{q}$:

$$\text{Peripheral-Attention}^{(h)}(\mathbf{X}, \mathbf{R})_{\mathbf{q},:} = \sum_{\mathbf{k} \in \mathcal{P}} \left( \frac{\Phi_{\text{c}}^{(h)}(\mathbf{X})_{\mathbf{q},\mathbf{k}} \cdot \Phi_{\text{p}}^{(h)}(\mathbf{R})_{\mathbf{q},\mathbf{k}}}{\sum_{\mathbf{j} \in \mathcal{P}} \Phi_{\text{c}}^{(h)}(\mathbf{X})_{\mathbf{q},\mathbf{j}} \cdot \Phi_{\text{p}}^{(h)}(\mathbf{R})_{\mathbf{q},\mathbf{j}}} \right) \mathbf{V}_{\mathbf{k},:}^{(h)} \tag{28}$$

$$= \sum_{\mathbf{k} \in \mathcal{P}} \left( \frac{\Phi_{\text{c}}^{(h)}(\mathbf{X})_{\mathbf{q},\mathbf{k}} \cdot \mathbb{1}[s(h) = \mathbf{R}_{\mathbf{q},\mathbf{k},:}]}{\sum_{\mathbf{j} \in \mathcal{P}} \Phi_{\text{c}}^{(h)}(\mathbf{X})_{\mathbf{q},\mathbf{j}} \cdot \mathbb{1}[s(h) = \mathbf{R}_{\mathbf{q},\mathbf{j},:}]} \right) (\mathbf{X}\mathbf{W}_{\text{val}}^{(h)})_{\mathbf{k},:} \tag{29}$$

$$= \sum_{\mathbf{k} \in \mathcal{P}} \left( \mathbb{1}[s(h) = \mathbf{R}_{\mathbf{q},\mathbf{k},:}] \right) (\mathbf{X}_{\mathbf{k},:} \cdot \mathbf{W}_{\text{val}}^{(h)}) \tag{30}$$

$$= \left( \sum_{\mathbf{k} \in \mathcal{P}} \mathbb{1}[s(h) = \mathbf{q} - \mathbf{k}] \cdot \mathbf{X}_{\mathbf{k},:} \right) \mathbf{W}_{\text{val}}^{(h)} \tag{31}$$

$$= \left( \sum_{\mathbf{k} \in \mathcal{P}} \mathbb{1}[\mathbf{k} = \mathbf{q} - s(h)] \cdot \mathbf{X}_{\mathbf{k},:} \right) \mathbf{W}_{\text{val}}^{(h)} \tag{32}$$

$$= \mathbf{X}_{\mathbf{q}-s(h),:} \, \mathbf{W}_{\text{val}}^{(h)} \tag{33}$$

$$\tag{34}$$

Assuming $D_h \geq D_{\text{emb}}$, the MPA is formulated as

$$\text{MPA}(\mathbf{X})_{\mathbf{q},:} = \sum_{h \in [N_h]} \left( \text{Peripheral-Attention}^{(h)}(\mathbf{X}, \mathbf{R})_{\mathbf{q},:} \right) \mathbf{W}_{\text{out}}^{(h)} + \mathbf{b}_{\text{out}} \tag{35}$$

$$= \sum_{h \in [N_h]} \left( \mathbf{X}_{\mathbf{q}-s(h),:} \mathbf{W}_{\text{val}}^{(h)} \right) \mathbf{W}_{\text{out}}^{(h)} + \mathbf{b}_{\text{out}} \tag{36}$$

$$= \sum_{h \in [N_h]} \left( \mathbf{X}_{\mathbf{q}-s(h),:} \mathbf{W}_{\text{val}}^{(h)} \mathbf{W}_{\text{out}}^{(h)} \right) + \mathbf{b}_{\text{out}} \tag{37}$$

$$= \sum_{h \in [N_h]} \left( \mathbf{X}_{\mathbf{q}-s(h),:} \mathbf{W}^{(h)} \right) + \mathbf{b}_{\text{out}} \tag{38}$$

$$= \sum_{\kappa \in \nabla_k} \mathbf{X}_{\mathbf{q}-\kappa,:} \mathbf{W}_{\kappa,:,:}^{(\text{conv})} + \mathbf{b}_{\text{out}} \tag{39}$$

$$= \text{Conv2D}\left( \mathbf{X}; \mathbf{W}^{(\text{conv})} \right)_{\mathbf{q},:} + \mathbf{b}_{\text{out}}, \tag{40}$$

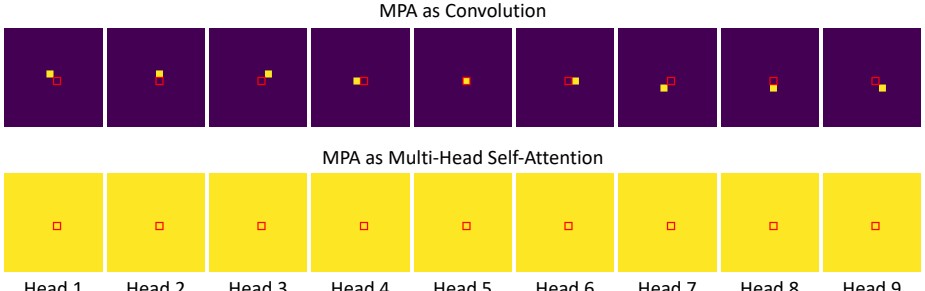

Figure S3: MPA layers with $N_h = 9$ heads in two extreme cases of semi-dynamic attention (top) and position bias injection (bottom) which respectively express $3 \times 3$ convolution and MHSA.

where $\mathbf{W}_{s(h)}^{(\text{conv})} := \mathbf{W}^{(h)} \in \mathbb{R}^{D_{\text{emb}} \times D_{\text{emb}}}$ is a weight matrix of 2-dimensional convolutional kernel, $\mathbf{W}^{(\text{conv})} \in \mathbb{R}^{k \times k \times D_{\text{emb}} \times D_{\text{out}}}$, at position $s(h)$[4]. ∎

**MPA as an multi-head self-attention.** Assume that the content-based attention is scaled dot product between queries and keys, *i.e.*, $\Phi_{\text{c}}^{(h)}(\mathbf{X}) = \exp(\tau \mathbf{Q}^{(h)}, \mathbf{K}^{(h)\top})$, and the bias term of the second instance normalization in $\Phi_{\text{p}}^{(h)}$ is set to some large number such that $\beta_{\text{p2}}^{(h)} = \infty$ for all heads $h$, *i.e.*, *an extreme (global) position bias injection*. Note that the latter assumption gives $\Phi_{\text{p}}^{(h)}(\mathbf{R})_{\mathbf{q},\mathbf{k}} = 1$ for all $\mathbf{q}, \mathbf{k} \in \mathcal{P}$ and $h \in [N_h]$ according to Eq. 20. Now consider Peripheral-Attention:

$$\text{Peripheral-Attention}^{(h)}(\mathbf{X}, \mathbf{R}) = \text{Normalize}\left[ \Phi_{\text{c}}^{(h)}(\mathbf{X}) \odot \Phi_{\text{p}}^{(h)}(\mathbf{R}) \right] \mathbf{V}^{(h)} \tag{41}$$

$$= \text{Normalize}\left[ \exp(\tau \mathbf{Q}^{(h)}, \mathbf{K}^{(h)\top}) \odot J_{HW,HW} \right] \mathbf{V}^{(h)} \tag{42}$$

$$= \text{softmax}\left( \tau \mathbf{Q}^{(h)}, \mathbf{K}^{(h)\top} \right) \mathbf{V}^{(h)} \tag{43}$$

$$= \text{Self-Attention}^{(h)}(\mathbf{X}), \tag{44}$$

where $J_{HW,HW} \in \mathbb{R}^{HW \times HW}$ refers to all-one matrix. ∎

Figure S3 illustrates MPA layers in two extreme cases of semi-dynamic attention and position bias injection with $N_h = k^2 = 9$.

---

[4]This proof is primarily based on the work of Cordonnier *et al.* [1].

# D  Additional Results and Analyses

In this section, we provide additional results and analyses of the proposed method.

**Nonlocality and impact measure comparisons with other baseline methods.** To investigate how the peripheral position encoding $\Phi_p$ benefits PerViT, we compare the nonlocality measure of PerViT-T with that of our baseline DeiT-T [18] and DeiT-T with state-of-the-art RPE method, *i.e.*, iRPE-K [24]. To measure the nonlocality $\Omega_a$ (Eq. 13 of our main paper) of the other models, we use $\Phi_a = \exp(\mathbf{Q}\mathbf{K}^\top) \in \mathbb{R}^{HW \times HW}$ for DeiT, and $\Phi_a = \exp(\mathbf{Q}\mathbf{K}^\top + \mathbf{R}_{\text{rpe-K}}) \in \mathbb{R}^{HW \times HW}$ for iRPE[5].

As seen in the left of Fig. S4, DeiT learns to attend more locally in early layers compared to the late ones but its overall nonlocality is higher than that of PerViT due to the absence of position information $\Phi_p$, implying that position encoding effectively encourages higher locality for fine-grained pattern recognition. Meanwhile, the iRPE effectively imposes locality on attentions but it provides similar magnitudes of nonlocality across different layers, *i.e.*, local attention at every layer, while those of PerViT and DeiT highly varies from early (local) to late (global) layers.

To demonstrate this phenomenon, we plot and compare the impacts of $\Phi_c$ and $\Phi_p$[6] on $\Phi_a$ in the middle and right of Fig. S4 and visualize learned position-based attention $\Phi_p$ of iRPE in Fig. S5. From the plots, we note that the impacts of content- and position-based attentions in iRPE also have relatively low variance compared to PerViT. We conjecture that RPE transformation layer in [24] only consists of a single linear projection while being shared across every layer in the network, thus providing attention maps with less diversity in terms of both shapes and sizes (Fig. S5 *vs.* Fig. S9).

Interestingly, we also observe that attended regions in iRPE complement each other to cover the whole visual field, quite similarly to PerViT. For example, the attended regions in the first and second heads of Layer 3 form horizontal attention in the *central region* while the third head attends to the rest in the *peripheral region*. Note that most layers (Layer 2-10) behave similarly as in *human peripheral vision*, which again support the main motivation of this work.

From this study, we draw following conclusions: (i) Sufficiently high locality imposed by position bias improves vision transformer. (ii) The position-based attentions should be in diverse shapes and sizes across different layers. (iii) Without any special supervisions, RPEs learn to model *peripheral vision* solely based on training images, proving that our work poses a promising direction.

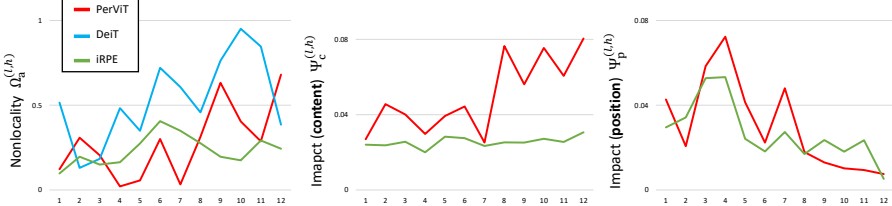

Figure S4: Nonlocality (left) and impact measure (middle: content-based $\Psi_c$, right: position-based $\Psi_p$) comparisons between DeiT [18], iRPE [24], and PerViT (ours).

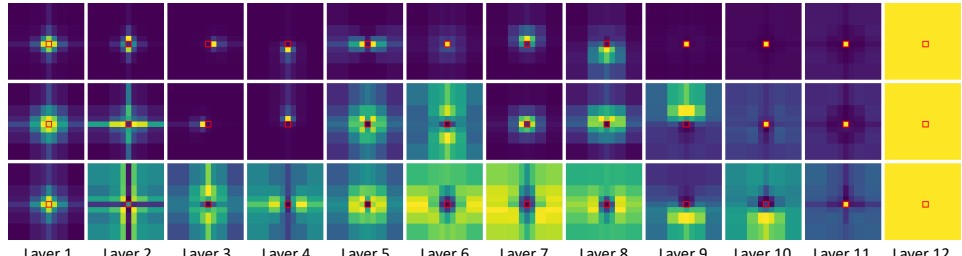

Figure S5: Learned position-based attention $\Phi_p$ of iRPE-K product method [24].

---

[5]The input to the exponential function is normalized to prevent large activations following the implementation of softmax in PyTorch [12]. For iRPE, we use contextual product method proposed in [24].

[6]We separate $\Phi_a = \exp(\mathbf{Q}\mathbf{K}^\top + \mathbf{R}_{\text{rpe-K}}) = \exp(\mathbf{Q}\mathbf{K}^\top) \odot \exp(\mathbf{R}_{\text{rpe-K}})$ in iRPE into two terms of content- and position-based attentions: $\Phi_c = \exp(\mathbf{Q}\mathbf{K}^\top)$ and $\Phi_p = \exp(\mathbf{R}_{\text{rpe-K}})$.

**Qualitative comparison between different network designs of $\Phi_p$.** In Tab. 4 of the main paper, we explored different network designs of $\Phi_p$ in PerViT-T: Euc (single-layer proj. w/o $\mathcal{N}$), Euc+ML (multi-layer proj. w/o $\mathcal{N}$), and Euc+$\mathcal{N}$+ML (multi-layer proj. with $\mathcal{N}$, *i.e.*, **ours**). We visualize their learned position-based attentions $\Phi_p$ in Fig. S6. As discussed in Sec. 3.1 and can be seen from first and third group of attentions, the single-layer projections are only able to provide Gaussian-like attention maps. We observe that referring neighbors $\mathcal{N}$ with single-layer severely damages performance; we suspect that unnecessarily large attention scores at distant positions (possibly caused by neighborhood aggregation) hinder the ability to focus on local patterns as seen from nonlocal circular attentions in the third group. The multi-layer (ML) projection in the second group helps the model in forming (torus-shaped) peripheral regions but it still poses undesirable rotational symmetric property in feature transformations. Note that the proposed ML projection with $\mathcal{N}$ (the last group of attentions) gently breaks the rotational symmetric property while retaining the ability to form peripheral regions to a sufficient extent with a significant performance boost, implying that modelling effective attention-based peripheral vision demands both designs of multi-layer and neighborhood.

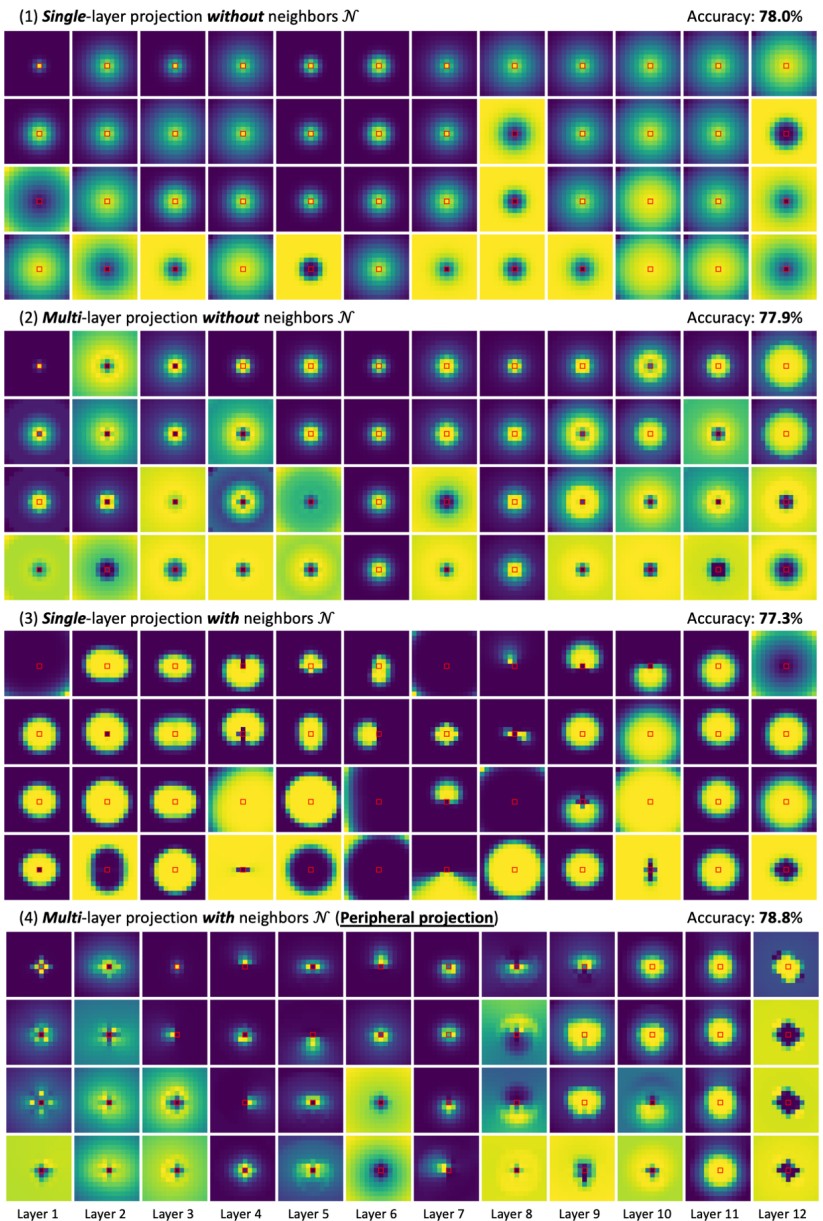

Figure S6: Learned position-based attention $\Phi_p$ with different network design choices.

**Further analyses on nonlocality** $\Omega_a$ **and** $\Omega_p$**.** In Tab. 1 of the main paper, we have analyzed how the absence of each component in PerViT ($\Phi_p$, $\Phi_c$, C-stem, and CPE) impacts on the performance, *e.g.*, ImageNet accuracy. In this study, we investigate their impact on *both performance and locality* in order to reveal their relationship in vision transformers.

As seen from Tab. S1[7], there exist noticeable performance gaps between the models (b, f, g, h) (without $\Phi_p$) and (a, d, e, i) (with $\Phi_p$). To see if the improvements are from the locality imposed by $\Phi_p$, we plot and compare the nonlocality measures of the models in Fig. S8. As seen from the top four plots, $\Phi_p$ always imposes early local attentions without any exceptions. The results prove the necessity of *local transformations in the early stages* and again support recent research directions towards augmenting early convolutions in vision transformer architectures [2, 8, 13, 14, 25].

The bottom-left plot of Fig. S8 reveals the impact of content-based attention $\Phi_c$ on attention locality. Without adaptive attention (model (c)), $\Phi_p$ imposes stronger locality on every layer. Compared to (a) (Fig. S9), (c) performs 'hard' attentions as seen from Fig. S7; especially, the early attentions look noticeably similar to convolutions as illustrated at the top of Fig. S3. Removing dynamicity encourages the model (c) to be more 'convolutional' as it loses the ability to adaptively collect relevant features over broader areas, exploiting only a few relevant ones in local regions statically.

Two plots at the bottom-right of Fig. S8 show the impact of convnets, *e.g.*, C-stem and CPE, on locality of mixed attention $\Phi_a$. We find that the absence of the convnets hardly affects the locality. In presence of $\Phi_p$, model performs early local and late global attention like PerViT (a). In absence of $\Phi_p$, the overall nonlocality increases for all models (b, f, g, i). The results reveal that built-in locality of convolutional layers does not have direct effects on locality of the self-attention layers.

Table S1: Ablation study.

| Ref. | $\Phi_p$ | $\Phi_c$ | C-stem | CPE | acc. |
|------|------|------|--------|-----|------|
| (a) | ✓ | ✓ | ✓ | ✓ | 78.8 |
| (b) | ✗ | ✓ | ✓ | ✓ | 77.3 |
| (c) | ✓ | ✗ | ✓ | ✓ | 76.8 |
| (d) | ✓ | ✓ | ✗ | ✓ | 77.8 |
| (e) | ✓ | ✓ | ✓ | ✗ | 78.1 |
| (f) | ✗ | ✓ | ✗ | ✓ | 76.3 |
| (g) | ✗ | ✓ | ✓ | ✗ | 76.7 |
| (h) | ✓ | ✓ | ✗ | ✗ | 76.5 |
| (i) | ✗ | ✓ | ✗ | ✗ | 72.3 |

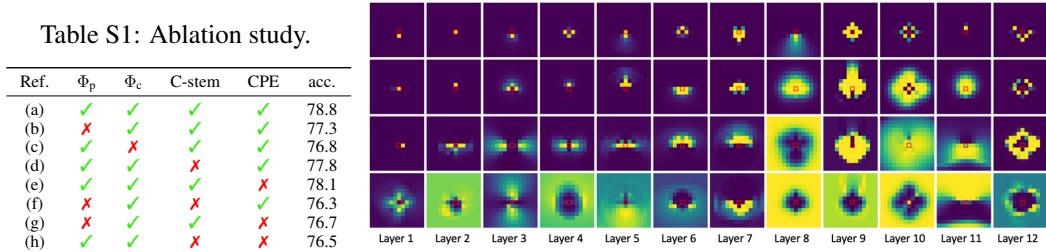

Figure S7: Learned position-based attention $\Phi_p$ of model (c).

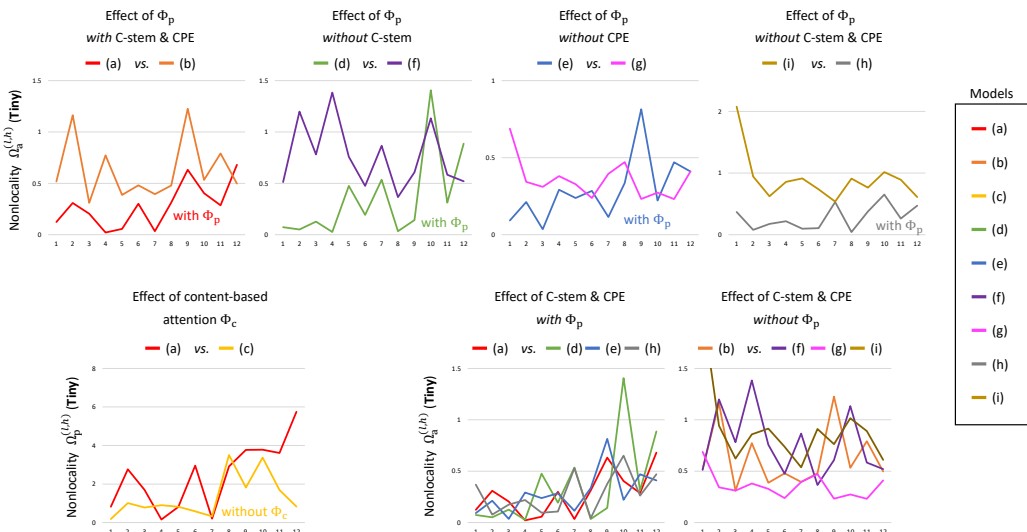

Figure S8: The effect of each component in PerViT to nonlocality measures $\Omega_a$ (top 4 and bottom-right 2 plots) and $\Omega_p$ (bottom-left plot).

---

[7]We copy the results of Tab. 1 in the main paper for the ease of demonstration.

**Visualization of learned position-based attention $\Phi_\mathbf{p}$.** Figures S9, S10, and S11 depict learned position-based attentions $\Phi_\mathbf{p}$ in PerViT-Tiny, Small, and Medium respectively where the attention maps at each layer are sorted in the order of nonlocality measure. We observe that the overall nonlocality of a model, *e.g.*, brightness of the attention maps, noticeably increases as the model size grows as discussed in Sec. 4.1 of the main paper. Interestingly, for all three models, the query position $\mathbf{q}$, *i.e.*, the center, is not attended in the most of the learned attention; we hypothesize that a sufficient amount of transformations on query position is already done by $3 \times 3$ depth-wise convolutions in CPE and point-wise $(1 \times 1)$ convolutions in MLP layers so query position no longer require further transformations in MPA which thereupon focuses mostly on *peripheral regions*. We also observe that MPA at Layer 5 of the Medium model performs uniform, global attentions at different scales just like MHSA as seen in Fig. S14.

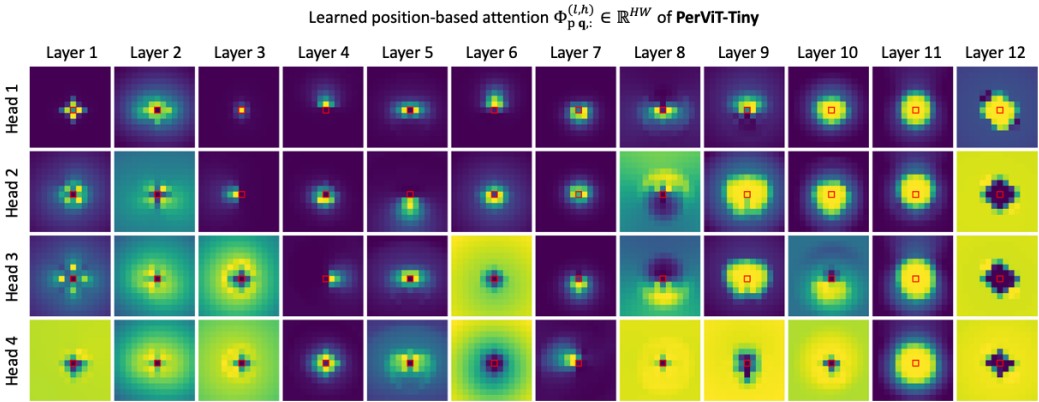

Figure S9: Learned position-based attentions $\Phi_\mathbf{p}{}^{(l,h)}_{\mathbf{q},:}$ of PerViT-Tiny.

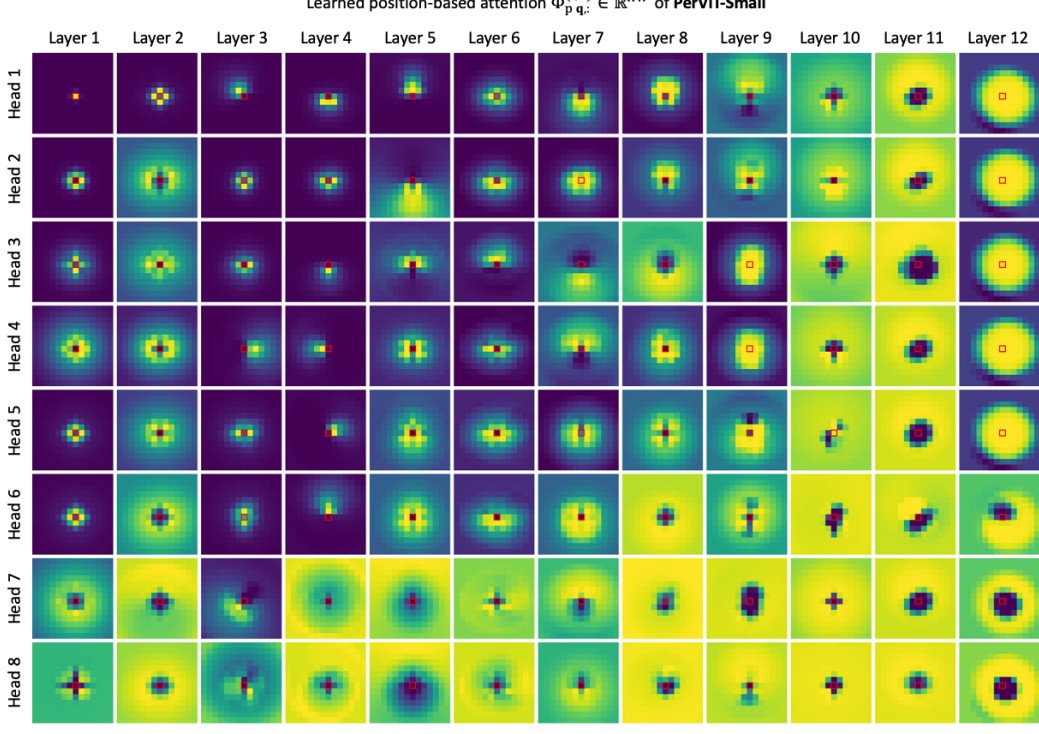

Figure S10: Learned position-based attentions $\Phi_\mathbf{p}{}^{(l,h)}_{\mathbf{q},:}$ of PerViT-Small.

Learned position-based attention $\Phi_{\mathrm{p}\,\mathbf{q},:}^{(l,h)} \in \mathbb{R}^{HW}$ of **PerViT-Medium**

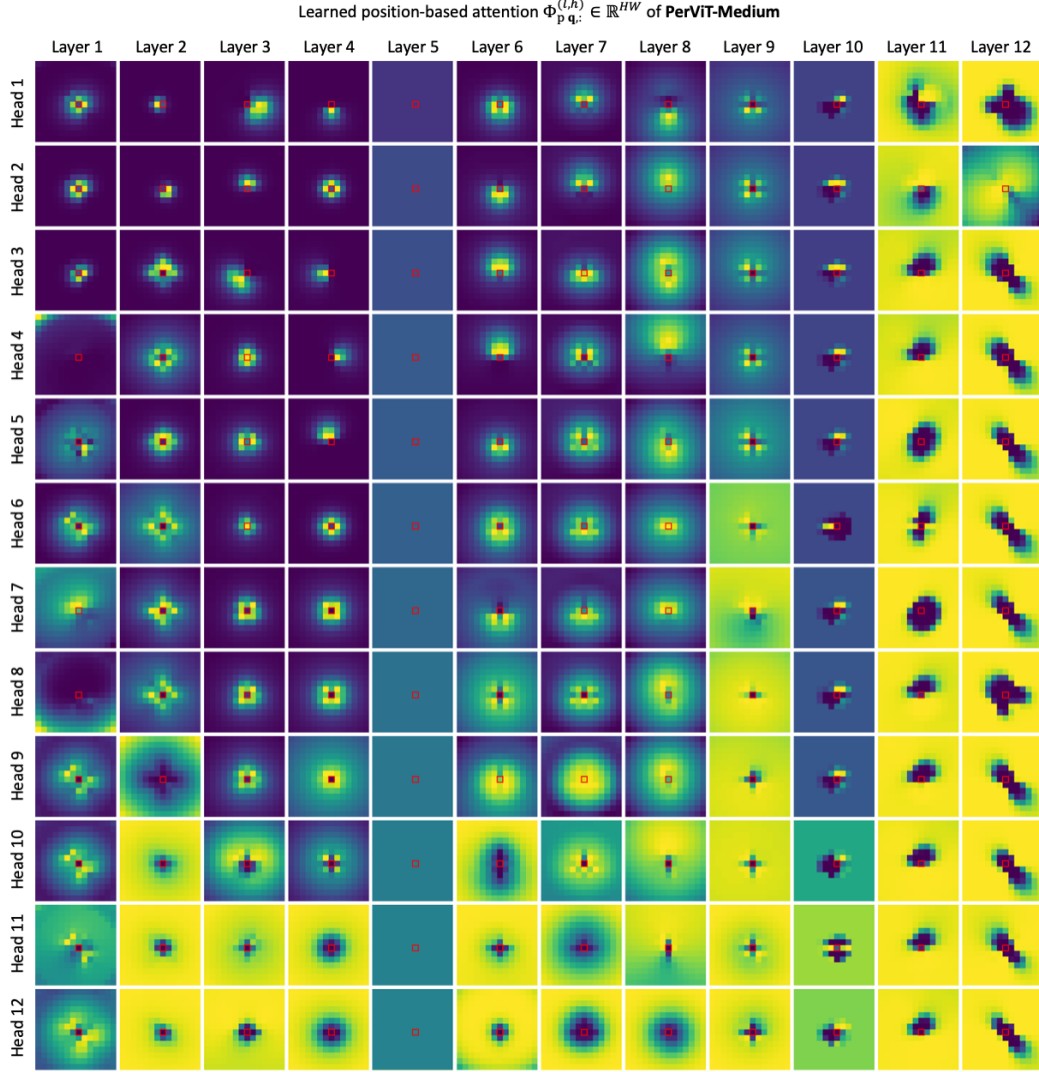

Figure S11: Learned position-based attentions $\Phi_{\mathrm{p}\,\mathbf{q},:}^{(l,h)}$ of PerViT-Medium.

**Further analyses on qualitative results.** We visualize sample attention maps of PerViT Tiny, Small, and Medium in Figs. S12, S13, and S14 respectively where position-based attentions $\Phi_p$ are shown at the top, and sample images with their mixed attentions $\Phi_a$ are listed below. We pick three layers among early to late layers to investigate how MPAs form mixed attention across different layers.

At early levels, *e.g.*, Layers 2-3, about half the number of heads in each MPA forms (semi-) static/local attentions in central regions while the others provide relatively dynamic/global attentions in the peripheral regions, all of which complement each other to cover whole visual field. Note that the attention scores in early to intermediate levels, *e.g.*, Layers 2-6, mostly fall inside the object of interest, capturing relevant visual patterns for effective image classification. However, at later level, *e.g.*, Layer 9, position-based attention scores $\Phi_p$ are formed in mid- and far-peripheral regions, and mixed attention scores $\Phi_a$ are scattered over the whole spatial area. We conjecture that the late layer MPAs collect complementary features which the network missed along early to intermediate feature processing pathways so as to reinforce the image embeddings for the final prediction.

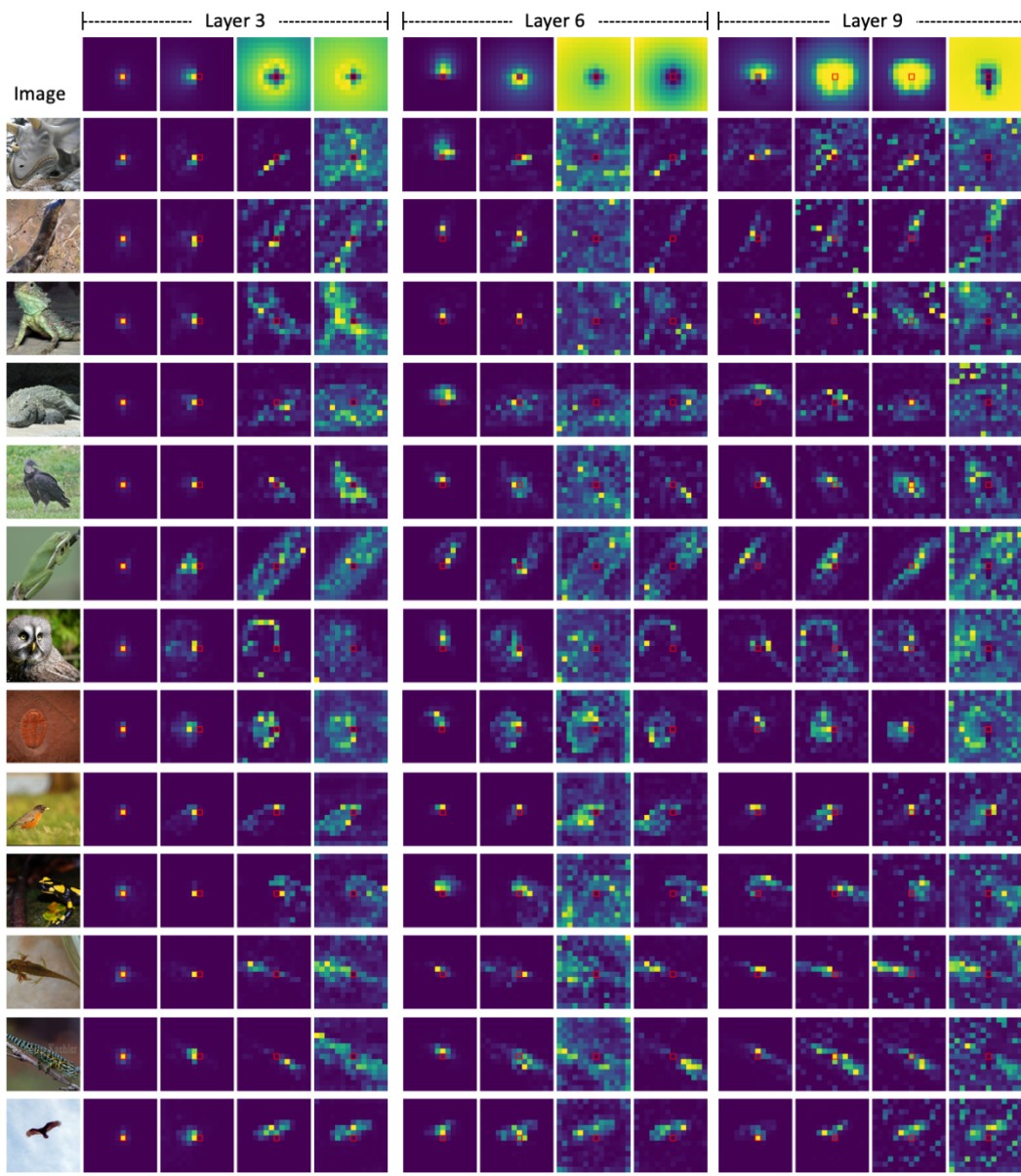

Figure S12: Visualization of attentions $\Phi_p^{(l,h)}$ and $\Phi_a^{(l,h)}$ of PerViT-Tiny.

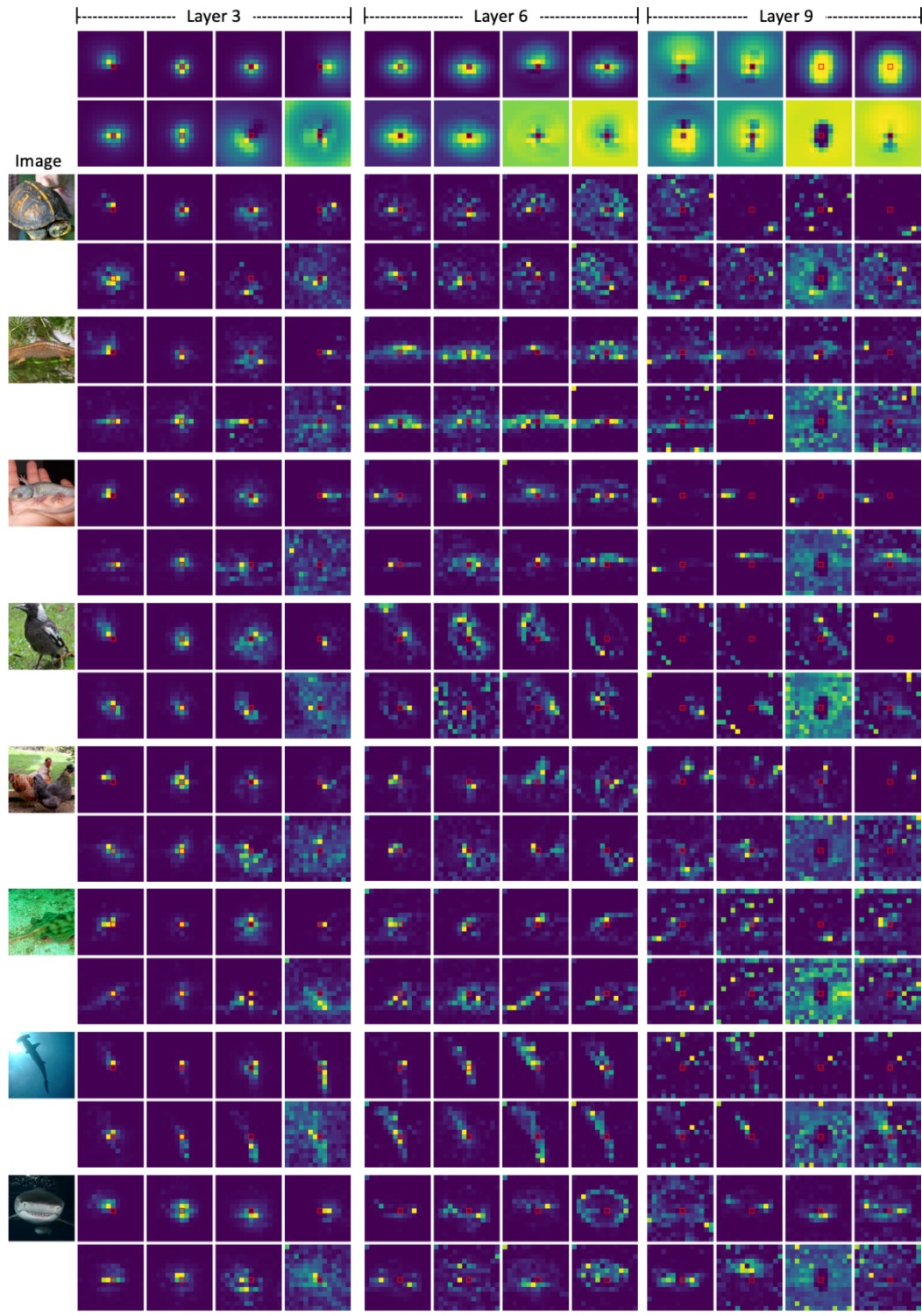

Figure S13: Visualization of attentions $\Phi_{\mathrm{p}}^{(l,h)}$ and $\Phi_{\mathrm{a}}^{(l,h)}$ of PerViT-Small.

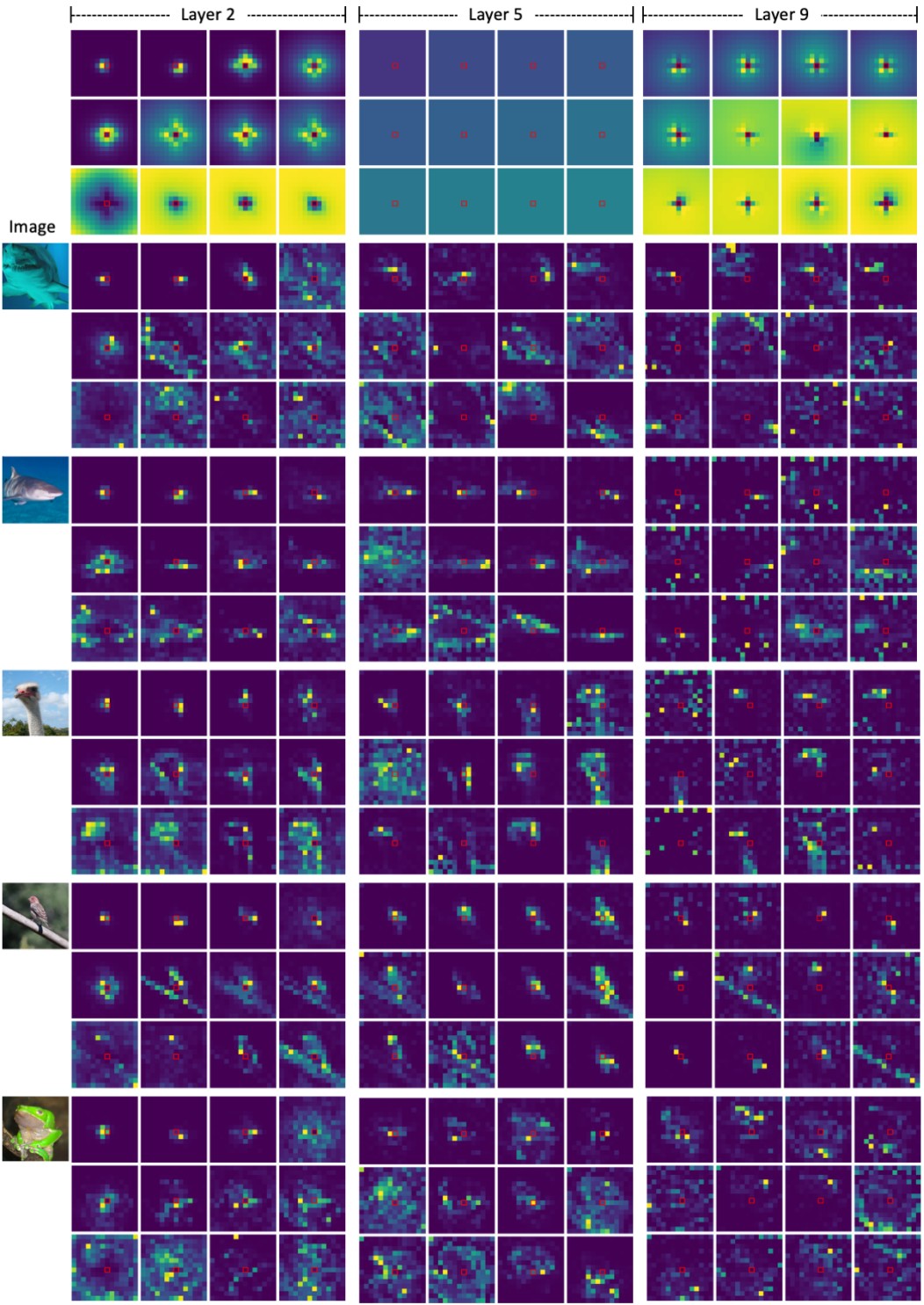

Figure S14: Visualization of attentions $\Phi_p^{(l,h)}$ and $\Phi_a^{(l,h)}$ of PerViT-Medium.

**The impact and nonlocality measures of** $\Phi_p^{(l,h)}$ **.** In the main paper, we provide the impact and nonlocality measures in bar graphs for Tiny model only due to the limited space. Here we provide the measures for all three different models of Tiny, Small, and Medium in Figs. S15 and S16.

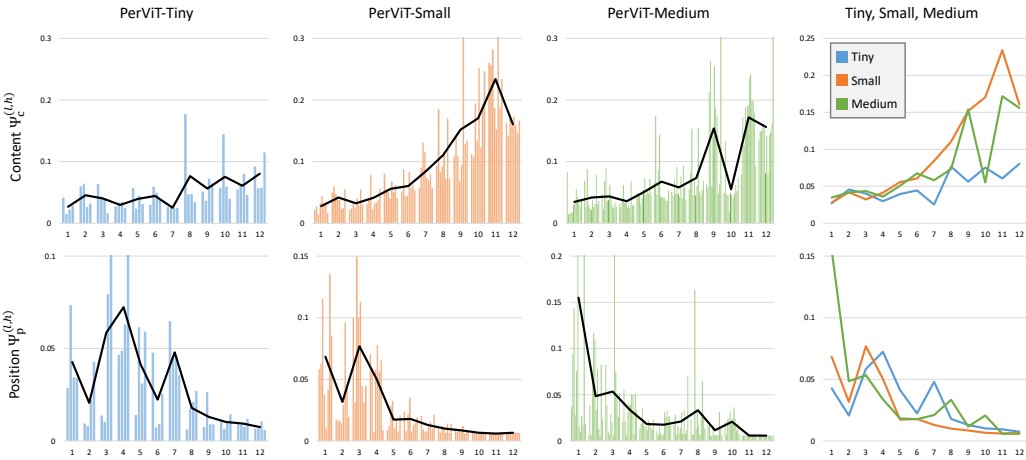

Figure S15: The measure of impact (x-axis: layer index, y-axis: the impact metric $\Psi_*$). Each bar graph shows the measure of a single head (# heads ($N_h$) are set to 4, 8, and 12 for Tiny, Small, and Medium models respectively), and the solid lines represent the trendlines which follow the average values of layers.

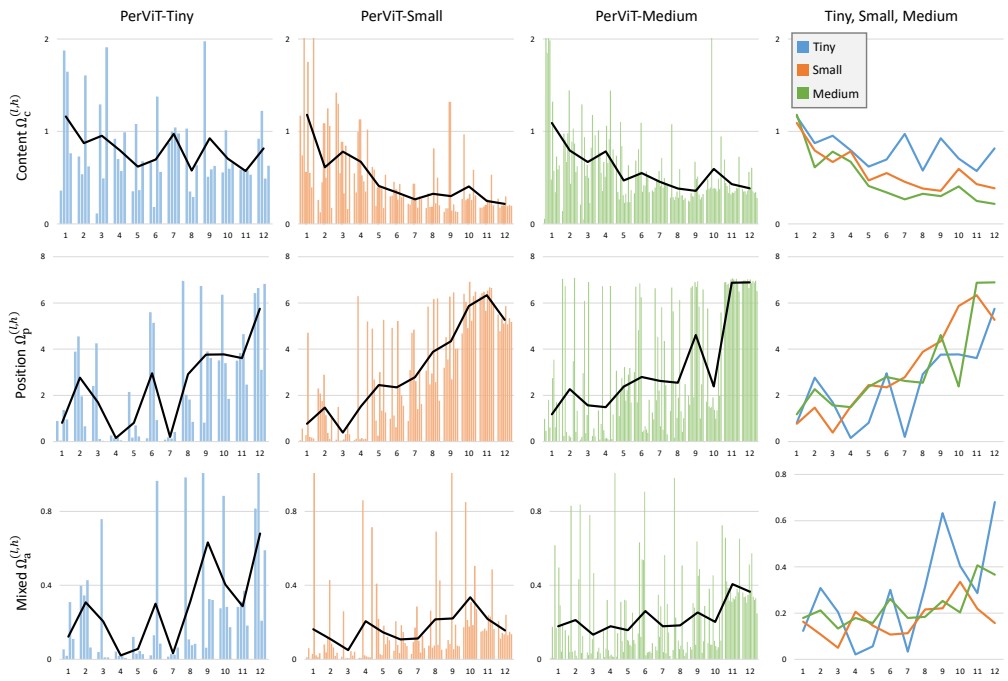

Figure S16: The measure of nonlocality (x-axis: layer index, y-axis: the nonlocality measure $\Omega*$).

**Comparison with additional state-of-the-art baselines.** Table S2 summarizes model sizes, computational costs, and ImageNet performances of recent state-of-the-art methods and ours. All the models are trained using images of $224 \times 224$ resolution. PerViT with different model sizes achieves better (Tiny) or highly competitive (Small and Medium) performances compared with convnet and pyramidal ViT counterparts. When compared with columnar vision transformers that use single-resolution feature maps, the proposed method performs the best.

Table S2: Performance comparison on ImageNet [4] with additional baselines.

|  | Model | Size (M) | FLOPs (G) | Top-1 (%) |
|---|---|---|---|---|
| **Fully Convolutional Networks** | ResNet-18 [7] | 12 | 1.8 | 69.8 |
| | RSB-ResNet-18 [21] | 12 | 1.8 | 70.6 |
| | ResNet-50 [7] | 25 | 4.1 | 78.5 |
| | RSB-ResNet-34 [21] | 22 | 3.7 | 75.5 |
| | RSB-ResNet-50 [21] | 26 | 4.1 | 79.8 |
| | ConvNext-T [11] | 29 | 4.5 | 82.1 |
| | ResNet-101 [7] | 45 | 7.9 | 79.8 |
| | RSB-ResNet-101 [21] | 45 | 7.9 | 81.3 |
| | RSB-ResNet-152 [21] | 60 | 12 | 81.8 |
| | ConvNext-S [11] | 50 | 8.7 | 83.1 |
| **Spatial MLP-Mixer** | ResMLP-S12 [17] | 15 | 3.0 | 76.6 |
| | gMLP-S [9] | 20 | 4.5 | 79.6 |
| | ResMLP-S24 [17] | 30 | 6.0 | 79.4 |
| | MLP-Mixer-B/16 [16] | 59 | 13 | 76.4 |
| | ResMLP-B24 [17] | 116 | 23 | 81.0 |
| | gMLP-B [9] | 73 | 16 | 81.6 |
| **Pyramidal Vision Transformers** *(multi-resolution)* | PVT-T [20] | 13 | 1.9 | 75.1 |
| | CoaT-Lite-T [26] | 5.7 | 1.6 | 77.5 |
| | PoolFormer-S12 [28] | 12 | 2.0 | 77.2 |
| | CvT-13 [23] | 20 | 4.5 | 81.6 |
| | PVT-S [20] | 25 | 3.8 | 79.8 |
| | Swin-T [10] | 28 | 4.5 | 81.3 |
| | CoaT-Lite-S [26] | 20 | 4.0 | 81.9 |
| | Focal-T [27] | 29 | 4.9 | 82.2 |
| | PoolFormer-S24 [28] | 21 | 3.6 | 80.3 |
| | PoolFormer-S36 [28] | 31 | 5.2 | 81.4 |
| | CvT-32 [23] | 32 | 7.1 | 82.5 |
| | PVT-M [20] | 44 | 6.7 | 81.2 |
| | CoAtNet-1 [2] | 42 | 8.4 | 83.3 |
| | Swin-S [10] | 50 | 8.7 | 83.0 |
| | Focal-S [27] | 51 | 9.1 | 83.5 |
| | CoaT-Lite-M [26] | 45 | 9.8 | 83.6 |
| | PoolFormer-M36 [28] | 56 | 9.1 | 82.1 |
| | PoolFormer-M48 [28] | 74 | 12 | 82.5 |
| **Columnar Vision Transformers** *(single-resolution)* | DeiT-T [18] | 5.7 | 1.3 | 72.2 |
| | ConViT-T [3] | 5.6 | 1.2 | 73.1 |
| | ViT$_C$-1GF [25] | 4.6 | 1.1 | 75.3 |
| | XCiT-T12/16 [6] | 7.0 | 1.2 | 77.1 |
| | **PerViT-T (ours)** | 7.6 | 1.6 | 78.8 |
| | DeiT-S [18] | 22 | 4.6 | 79.8 |
| | ConViT-S [3] | 22 | 5.4 | 81.3 |
| | ViT$_C$-4GF [25] | 18 | 4.0 | 81.4 |
| | T2T-ViT$_t$-14 [29] | 22 | 6.1 | 81.7 |
| | XCiT-S12/16 [6] | 26 | 4.8 | 82.0 |
| | **PerViT-S (ours)** | 21 | 4.4 | 82.1 |
| | DeiT-B [18] | 86 | 18 | 81.8 |
| | ConViT-S+ [3] | 48 | 10 | 82.2 |
| | T2T-ViT$_t$-24 [29] | 64 | 15 | 82.6 |
| | XCiT-S24/16 [6] | 48 | 9.1 | 82.6 |
| | **PerViT-M (ours)** | 44 | 9.0 | 82.9 |

Table S3: Ablation on $\Phi_p$ under different initializations.

| Param. of $\Phi_p$ | Peripheral (ours) | Conv | Rand |
|---|---|---|---|
| Trained | **78.8** | 78.6 | 78.5 |
| Fixed | **77.8** | 77.1 | 75.8 |
| Absent | | 77.3 | |

Table S4: Transfer learning results on CIFAR-10, CIFAR-100, and iNaturalist-19.

| Model | Size (M) | FLOPs (G) | CIFAR-10 | CIFAR-100 | iNAT-19 |
|---|---|---|---|---|---|
| ViT-L [5] | 307 | 117 | 97.9 | 86.4 | - |
| DeiT-B [18] | 86 | 18 | **99.1** | 90.8 | 77.7 |
| PerViT-M (ours) | **44** | **9** | **99.1** | **91.4** | **78.5** |

**Additional ablation on $\Phi_p$.** To highlight the benefits of learning $\Phi_p$, we conduct experiments using PerViT-T with parameters of $\Phi_p$ fixed during training under three different initialization methods of peripheral, conv, and rand. Table S3 summarizes the results. Fixing $\Phi_p$ parameters damages performance for all three intializations, verifying the efficacy of learning diverse position-based attentions across different layers and heads. We observe that conv and rand inits perform poorly compared to PerViT-T without $\Phi_p$, *e.g.*, model (b) in Tab. 1(77.3%); we suspect that fixed $\Phi_p$ with conv and rand inits only provide local and noisy attentions respectively while PerViT without $\Phi_p$ has no such strong restrictions.

**Transfer learning results.** We verify the robustness of the proposed method by comparing the PerViT-M with baseline models [5, 18] on different transfer learning task with ImageNet pre-training in Tab. S4. We finetune trained PerViT-M on CIFAR-10, CIFAR-100, and iNaturalist-19, following the same training recipes of DeiT [18]. Even with significantly lower complexity than [5, 18], our method surpasses baselines by approximately 1%p on CIFAR-100 and iNaturalist19 while performing on par with [18] on CIFAR-10.

# E  Model Layout Details

This paper explores PerViT with three different model sizes: Tiny, Small, and Medium. Each model consists total 12 blocks of layers ($N_l = 12$) divided into 4 stages each of which uses different channel sizes ($D_{emb} = D_h N_h$). The expansion ratio of the MLP layer is set to 4 following DeiT [18]. We use normalized coordinates to ensure numerical stability during training such that $[-1, -1]^\top \leq \mathbf{q}, \mathbf{k} \leq [1, 1]^\top$. The input and hidden dimensions of peripheral projections are set as $D_r = D_{hid} = 4N_h$. We set $K = 3$ to capture neighboring distance representations ($\mathcal{N}$) in the peripheral projections; in experiments, increasing the neighborhood size ($K > 3$) hardly brought improvements. We hypothesize the $3 \times 3$ window is sufficient for the model to capture the spatial dimension of the input images. Similarly to our baseline, *e.g.*, DeiT [18], the largest model (PerViT-M) require a few days of training on 8 V100 GPUs.

Table S5 summarizes architecture details for the three models. As seen in the last two columns, despite the small size of the proposed peripheral position encoding $\Phi_p$ ($\leq 0.6\%$ of the network size), it significantly boosts top-1 ImageNet accuracy (1.4%p~4.2%p improvements with $\Phi_p$ for Tiny) as shown in Tab. 1 of the main paper, verifying its effectiveness in terms of both accuracy and efficiency.

Table S5: Model layout for PerViT Tiny (T), Small (S), and Medium (M) models.

| PerViT model | # heads ($N_h$) | # layers at each stage | Channel sizes at each stage ($D_{emb}$) | Channel sizes in conv. patch embedding | Size (M) | FLOPs (G) | Attention $\Phi_p$ Size (M) | Ratio |
|---|---|---|---|---|---|---|---|---|
| Tiny | 4 | [2, 2, 6, 2] | [128, 192, 224, 280] | [48, 64, 96, 128] | 7.6 | 1.6 | 0.04M | 0.3% |
| Small | 8 | [2, 2, 6, 2] | [272, 320, 368, 464] | [64, 128, 192, 262] | 21.3 | 4.4 | 0.14M | 0.6% |
| Medium | 12 | [2, 2, 6, 2] | [312, 468, 540, 684] | [64, 192, 256, 312] | 43.7 | 9.0 | 0.31M | 0.6% |

## F  Training Details

We follow similar training recipes of DeiT [18], a hyperparameter-optimized version of ViT [5]. The stochastic depth rate is used in Small and Medium models where we set them to 0.1 and 0.2 respectively. For Medium model, we use 20 warmup epochs as smaller warmup epoch did not converge in our experiments. Table S6 summarizes the details on our training recipe.

Table S6: Training parameters for PerViT Tiny (T), Small (S), and Medium (M) and DeiT-B [18].

| Methods | | PerViT | | DeiT-B [18] |
|---|---|---|---|---|
| Epochs | | 300 | | 300 |
| Batch size | | 1024 | | 1024 |
| Optimizer | | AdamW | | AdamW |
| Learning rate | | $0.0005 \times \frac{batchsize}{512}$ | | $0.0005 \times \frac{batchsize}{512}$ |
| Learning rate decay | | cosine | | cosine |
| Weight decay | 0.03 (T) | 0.05 (S) | 0.05 (M) | 0.05 |
| Warmup epochs | 5 (T) | 5 (S) | 20 (M) | 5 |
| Label smoothing $\epsilon$ | | 0.1 | | 0.1 |
| Dropout | | ✗ | | ✗ |
| Stoch. Depth | 0.0 (T) | 0.1 (S) | 0.2 (M) | 0.1 |
| Repeated Aug | | ✓ | | ✓ |
| Gradient clip | | ✗ | | ✗ |
| Rand Augment | | 9/0.5 | | 9/0.5 |
| Mixup prob. | | 0.8 | | 0.8 |
| Cutmix prob. | | 1.0 | | 1.0 |
| Erasing prob. | | 0.25 | | 0.25 |

## G  Societal Implications and Broader Impacts

The focus of this work is model exploration & development for image classification task, providing an original direction towards combining human peripheral vision with transformer-based architecture. To the best of our knowledge, this work poses no immediate negative impact on society other than environmental concerns related to $CO_2$ emisssion; training vision transformers on large-scale dataset [4] from scratch demands a substantial amount of computational resources, *e.g.*, GPUs.

Our work may inspire biologically-inspired computer vision researches, which would potentially promote the creation of stronger machine vision. While we have focused on image classification task, we believe that the idea can be broadly applicable for high-level vision applications such as object detection & segmentation, and action recognition. We leave this to future work.