# OpenReview forum: "Peripheral Vision Transformer"
_NeurIPS.cc/2022/Conference — NeurIPS 2022 Accept_

### Official Review · Reviewer_5MyL · 2022-07-10

**Rating:** 6
**Confidence:** 3
**Soundness:** 3 good
**Presentation:** 2 fair
**Contribution:** 2 fair

**Summary:**

In this paper, the authors introduce a novel transformer biologically inspired transformer architecture. They introduce a change on the attention mechanism which enable models to split the visual field into different peripheral regions. Authors compare their proposal to the baseline on classification, showing improvement.


**Questions:**

My concerns are listed in the weaknesses section. My main questions are:

- Have the authors considered other tasks/datasets for evaluation? In my opinion the evaluation is very limited.

- How would the authors argue that the paper is still contributing to pushing the SOTA if Focal-S performs better than the proposed model.

- Have the authors evaluated the quality of the segmentation in segmentation data?

**Limitations:**

Yes, I think the discussion of limitations is correct.

**Strengths And Weaknesses:**

**Strengths**:
- S1. The paper proposes an interesting and useful addition to standard transformer attention. State-of-the-art methods typically lack of the ability to partition the visual field in peripheral regions, which can help focusing on the most important content for the model and make training more efficient.

- S2. The paper is well written and motivated. I think authors described well the goal of the paper and the model, as well as illustrated well the idea with plenty of useful figures.

- S3.  The ablation study in the paper is very complete, going over all posible design decisions that shaped the final architecture and model. I specially like the results in Table 1, where the reader can easily understand the importance of the different elements.

** Weaknesses**:
In my opinion the main weakness of the paper is the evaluation. I think authors fail to show that their proposed method is better than other proposals in the transformer area. My main concerns are:

- W1. Authors only evaluate on one ImageNet, and do not provide additional results on other datasets. In my opinion for the paper to be convincing about the usefulness of the methods, author should provide results on other datasets as well.

- W2. Focal-S performs better than the proposed model with similar number of flops and model size. As the ImageNet evaluation is the only data point we have to evaluate the model performance, it's hard to convince the reader with this evidence that the model is pushing the state-of-the-art.

- W3. Given the main innovation of the model, I believe that an object centric dataset such as ImageNet is probably not the best dataset for evaluation. I think the proposed model is good at focusing to the different objects in the image in an efficient manner, while in ImageNet typically a single object is present.

- W4. Following up with W3, I think this model could actually shine more on tasks such as segmentation, where fine detail of the input image is important to produce a quality output.

- W5. Have the authors considered evaluating the segmentation masks generated by the attention? (Figure 6)



------------------------------
* After rebuttal*: The author's response addressed most of my concerns and I updated my review score recommending acceptance.

---

> ### Author Response · Authors · 2022-08-01
> **Author response to reviewer 5MyL**
>
> We thank reviewer 5MyL for constructive comments and suggestions and will revise our paper by reflecting them as much as possible.
>
> **[Experiments on other datasets]**
> To further verify the robustness of the proposed method, we compare the PerViT-M with baseline models [11, 15] on different transfer learning task with ImageNet pre-training in **Tab.R6**. We finetune trained PerViT-M on CIFAR-10, CIFAR-100, and iNaturalist-19, following the same training recipes of DeiT [11]. Even with significantly lower complexity than  [11, 15], our method surpasses baselines by approx. 1%p on CIFAR-100 and iNaturalist19 while performing on par with [11] on CIFAR-10. Please excuse the absence of results on Flowers, Cars, and iNaturalist-18 datasets due to the tight submission deadline of the rebuttal. We will do our best to include them all in our final manuscript.
>
> **Table R6**. Transfer learning results on CIFAR-10, CIFAR-100, and iNAT-19.
> |      Model      | Size | GFLOPs | CIFAR-10 | CIFAR-100 | iNAT-19 | ImageNet-1K |
> |:---------------:|:----:|:------:|:--------:|:---------:|:-------:|:-----------:|
> |  ViT-L/16 [15]  |  307 |   117  |   97.9   |    86.4   |    -    |     76.5    |
> |   DeiT-B [11]   |  86  |   18   |   99.1   |    90.8   |   77.7  |     81.8    |
> | PerViT-M (ours) |  44  |    9   |   99.1   |    91.4   |   78.5  |     82.9    |
>
> **[The claim of state of the art]**
> We agree on the reviewer’s point that our claim of state of the art in the current manuscript needs to be revised. We will revise the claim in L11 and L69 accordingly as follows: “The performance improvements in image classification task over the columnar Transformer baselines, e.g., DeiT, across different model sizes demonstrate the efficacy of the proposed method.”
>
>
> **[Segmentation results]**
> We truly appreciate the reviewer’s motivating feedback that our method could shine more on downstream tasks like semantic segmentation. However, please excuse the absence of the segmentation results in this rebuttal due to its tight submission deadline; the implementation and experiments demanded more time than we expected. Nevertheless, we will do our best to include the requested segmentation results during the author-reviewer discussion period to further improve our manuscript. Meanwhile, we’d like to note that some work [16, 31, 43, 48, 53, 54] indeed report segmentation results but most of the relevant work [6, 8, 9, 11, 15, 37, 42, 45, 49, 51, 55, 57] did not include them, focusing on in-depth analyses and ablations of their methods from theoretical perspectives. In similar manner, a significant part of our current draft focuses on the systematic investigation on the inner workings of PerViT, e.g., how it learns to model peripheral vision, the impact of attentions, the role of locality, exploration of possible design choices of $\Phi_p$, and rigorous proof that makes our results reasonable and interpretable, all of which encourage an inspiring direction in computer vision literature.

---

> > ### Author Response · Authors · 2022-08-09
> > **Author response to reviewer 5MyL (continued)**
> >
> > **[Segmentation results]**
> > In the rebuttal, we promised to evaluate our model on other downstream tasks such as object detection and semantic segmentation. However, we found that the quadratic complexity of our model makes the evaluations on time infeasible; these downstream tasks typically take input images of higher resolutions (approx. $800 \times 1000$ images) than classification, being usually performed only by the networks that process inputs within feasible memory and time complexity, e.g, linear complexity with respect to the number of input tokens as in XCiT [16]. For example, some ViT-based work in Tab.3 [16, 23, 31, 54] introduce efficient self-attention techniques such as cross-covariance attention [16], squeezed convolutional projection [23], and window-wise attentions [31, 53], thus being able to evaluate their models on detection/segmentation. Meanwhile, the relevant work of  [3, 15, 18, 25, 55] (as well as ours) with ‘quadratic complexity’ do not perform such evaluations as they typically generate out-of-memory (OOM) errors given high-resolution input images. Nevertheless, to address the reviewer’s concern, we explored a number of variants of PerViT during the rebuttal to provide the reviewers with ‘memory-efficient PerViT’ but this exploration is constrained by the limited time of the rebuttal; the ideation & exploration of memory-efficient $\Phi_p$, the implementation, pre-training on ImageNet-1K, downstream task evaluation all demand a fair amount of time and resources for the evaluations to be complete. Although we were not able to fully address the concerns, we appreciate the reviewer’s feedback that helped us discover the reason why the current implementation of PerViT is limited for other tasks of high-resolution inputs. To make our manuscript more self-contained, we will discuss this point in the limitation section (Sec.5) in our final manuscript.

---

> > ### Comment · Reviewer_5MyL · 2022-08-09
> > **Thanks!**
> >
> > Dear authors,
> >
> > Thank you for the detailed and informative rebuttal. After reading the other reviews and the additional results in the rebuttal my most relevant concerns are resolved.
> >
> > Best wishes.

---

> > > ### Author Response · Authors · 2022-08-09
> > > **Reply to reviewer 5MyL**
> > >
> > > We again thank the reviewer for the motivating feedback and are glad to hear that reviewer 5MyL's most relevant concerns are addressed by our rebuttal. We find, however, the rating remains as before and thus will be truly appreciated if the recommendation is updated. Thank you in advance!

---

> > > > ### Comment · Reviewer_5MyL · 2022-08-09
> > > > **Reply to authors**
> > > >
> > > > Thank you authors for your message and flagging the missing score change. I just upgraded my score to reflect the changes.
> > > >
> > > > Best wishes.

---

> > > > > ### Author Response · Authors · 2022-08-09
> > > > > **Reply to reviewer 5MyL**
> > > > >
> > > > > We truly appreciate the positive evaluations and will do our best to reflect the comments as much as possible.

---

### Official Review · Reviewer_XfL6 · 2022-07-10

**Rating:** 7
**Confidence:** 4
**Soundness:** 3 good
**Presentation:** 3 good
**Contribution:** 3 good

**Summary:**

This paper proposed a novel visual transformer model that incorporates the inductive bias of peripheral vision into the relative positional encoding. Specifically, the authors define a multi-head peripheral attention (MPA) module, where content-based and position-based attention scores are first calculated independently and then combined through an element-wise product. The design of the position-based attention allows pairs of locations ($q$,$k$) with the same Euclidean distance to have similar attention scores. The peripheral projection introduces the diversity, while the peripheral initialization ensembles the increasing spatial receptive fields. The results show better performance on ImageNet classification task than models with similar capacity. The ablation studies suggest the major contribution of the performance improvement comes from the position-based peripheral attention in the MPA.

**Questions:**

Major:
- The "peripheral projections" part in Section 3.1 seems interesting and important, but it's hard to fully understand its intuition and computation from the description. Does this peripheral projection allow the attention to be calculated not from the key location itself, but from a small surrounding area (local context) around the key location? Does $PP$ in equation 8 stand for the computation of $\Phi_p$ in equation 7? It will be helpful if the authors may rephrase this part in a clearer and more intuitive way, or maybe use a visual illustration if it's helpful. I want to understand the "peripheral projection" better since it has the most contribution according to the results in Table 4.
- How is the $\mathcal{N}(\cdot)$ function defined in equation 7? Is it fixed for different layers/heads?
- Why there is a vertical discontinuity in the attention patterns after peripheral projections (Fig. 2, middle row)?
- From Table 4, it seems that adding ML (multi-layer design) to Euc doesn't help but adding $\mathcal{N}$ (peripheral projection) to Euc+ML improves the performance significantly. Is the multi-layer design an essential component or do we only need the diversity from $\mathcal{N}$?


Minor:
- The tables are not referenced in the order as they appear
- Some discussion of how peripheral vision affects model behavior in a human-like way may be inspiring when sketching future directions.
- It may be worth including a few early works that design biologically-inspired peripheral vision models with CNNs by polar transformation or using non-uniform sampling to generate foveated images, for example:
    1. Wang, P., & Cottrell, G. W. (2017). Central and peripheral vision for scene recognition: A neurocomputational modeling exploration. Journal of vision, 17(4), 9-9.
    2. Reddy, M. V., Banburski, A., Pant, N., & Poggio, T. (2020). Biologically inspired mechanisms for adversarial robustness. arXiv preprint arXiv:2006
    3. Jonnalagadda, A., Wang, W., & Eckstein, M. P. (2021). Foveater: Foveated transformer for image classification. arXiv preprint arXiv:2105.14173.



**Limitations:**

No negative societal impact.

**Strengths And Weaknesses:**

Strength:
- This paper proposed a concise idea to incorporate peripheral vision into the transformer-based computer vision models. The definition of relative positional encoding is straightforward to apply and understand and is well-aligned with the distance-based sampling in the human visual system.
- The performance evaluation and ablation studies convincingly suggest that peripheral attention is the critical module that contributes to the improvement, and it has a reasonable and interpretable behavior (e.g., the position bias is mainly seen in early layers; larger models have farther peripheral information).

Weakness:
- The evaluation of ImageNet classification may not be the best task to show the benefit of adding a peripheral mechanism to the computer vision model. It would be great if the authors could demonstrate if this mechanism significantly increases the efficiency and robustness of the model.
- The presentation of the methods and results can be improved for clarity.

---

> ### Author Response · Authors · 2022-07-30
> **Author response to reviewer XfL6**
>
> We thank reviewer XfL6 for insightful comments and positive evaluation on our work. We will revise the paper by reflecting them as much as possible.
>
> **[Experiments on other tasks & datasets]**
> We truly appreciate the reviewer’s motivating feedback that our method could benefit computer vision models in different tasks like semantic segmentation which is suggested by reviewer 5MyL as well. However, please excuse the absence of results on other tasks in this rebuttal due to its tight submission deadline; the implementation and experiments for the task of segmentation demanded more time than we expected. Nevertheless, we will do our best to include the requested results during the author-reviewer discussion period to further improve our manuscript. Instead, we evaluate our model, e.g., PerViT-M, on different transfer learning task with ImageNet pre-training and compare the results with baseline models of [11, 15] in **Tab.R6**. We finetune trained PerViT-M on CIFAR-10, CIFAR-100, and iNaturalist-19, following the same training recipes of DeiT [11]. Even with significantly lower complexity than  [11, 15], our method surpasses baselines by approx. 1%p on CIFAR-100 and iNaturalist19 while performing on par with [11] on CIFAR-10. Please excuse the absence of results on Flowers, Cars, and iNaturalist-18 datasets due to the tight submission deadline of the rebuttal. We will do our best to include them all in our final manuscript.
>
> **Table R6**. Transfer learning results on CIFAR-10, CIFAR-100, and iNAT-19.
> |      Model      | Size | GFLOPs | CIFAR-10 | CIFAR-100 | iNAT-19 | ImageNet-1K |
> |:---------------:|:----:|:------:|:--------:|:---------:|:-------:|:-----------:|
> |  ViT-L/16 [15]  |  307 |   117  |   97.9   |    86.4   |    -    |     76.5    |
> |   DeiT-B [11]   |  86  |   18   |   99.1   |    90.8   |   77.7  |     81.8    |
> | PerViT-M (ours) |  44  |    9   |   99.1   |    91.4   |   78.5  |     82.9    |
>
>
>
>
>
> **[Clarification on our method, e.g., peripheral projections]**
> The proposed position-based attention $\Phi_{p}$ extends the idea of previous relative positional (RP) encoding work [10, 15, 22, 37, 38, 50] which all use a single-layer linear projection, i.e., neural networks. In our work, we adopt a multi-layered design to provide torus-shaped attentions (top-right of Fig.2) and peripheral projections (PP) to break rotational symmetric properties (middle of Fig.2) to model peripheral vision. Specifically, given relative positions as inputs $\mathbf{R} \in \mathbb{R}^{HW \times HW \times D_r}$, PP with $\mathbf{W} \in \mathbb{R}^{K^2 \times D_r \times D_{\text{hid}}}$ is formally defined as follows: $PP(\mathbf{R}, \mathbf{W})_{\mathbf{q}, \mathbf{k}, :} \coloneqq \sum\_{\mathbf{m} \in \mathcal{N}(\mathbf{k})} \mathbf{R}\_{\mathbf{q}, \mathbf{m}, :} \mathbf{W}\_{\mathbf{m} - \mathbf{k}, :, :}$ where the neighbor function $\mathcal{N}$ provides a set of neighbors around given input key position $\mathbf{k}$ and is formally defined as $\mathcal{N}(\mathbf{k}) \coloneqq \left[ \mathbf{k}-[\frac{K}{2}], \dots, \mathbf{k}+[\frac{K}{2}] \right] \times \left[ \mathbf{k}-[\frac{K}{2}], \dots, \mathbf{k}+[\frac{K}{2}] \right]$. We use $K=3$ for all layers and heads of PerViT in our experiments as $K > 3$ hardly brought improvements (Supp. L201). We will rephrase this part (with a visual illustration if possible) in our final manuscript.
>
> **[Answers to questions]**
>
> - Peripheral projections: please refer to our response above.
>
> - $\mathcal{N}(\cdot)$: please refer to our response above.
>
> - Discontinuity in the learned attention maps: In page 8 of supp., we explore different network designs of $\Phi_p$ and perform qualitative comparisons. Comparing (1, 2) with (3, 4) of Fig.S6, we observe that $3 \times 3$ kernel in $\mathcal{N}(\cdot)$ creates vertical/horizontal discontinuities in attentions, capturing vertical/horizontal relationship of visual features while the multi-layer design gives more diversity in shapes.
>
> - The necessity of ML and $\mathcal{N}$ for $\Phi_p$: As demonstrated in page 8 of supp., modeling effective peripheral vision demands both designs of ML and $\mathcal{N}$; ML helps the model in forming (torus-shaped) peripheral regions while $\mathcal{N}$ gently breaks the rotational symmetric property (L135-139).

---

> > ### Author Response · Authors · 2022-08-02
> > **Author response to reviewer XfL6 (continued)**
> >
> >
> > **[Minor points]**
> > We thank the reviewer for detailed feedback and will revise them all accordingly.
> >
> > **[Discussion on biologically-inspired models]**
> > We appreciate the suggested references [F, G, H]. Following suggestions from the reviewers XfL6 and mNPH, we will extend the related work section to include recent literature on peripheral vision, biologically-inspired machine vision methods [B-H], and a discussion of how peripheral vision affects model behavior in a human-like way.
> >
> >
> >
> >
> >
> > [A] He et al. (2022). Masked Autoencoders Are Scalable Vision Learners.
> >
> > [B] Rosenholtz R. (2016). Capabilities and Limitations of Peripheral Vision.
> >
> > [C] Balas et al. (2009). A summary-statistic representation in peripheral vision explains visual crowding.
> >
> > [D] Deza et al. (2020). Emergent Properties of Foveated Perceptual Systems.
> >
> > [E] Deza et al. (2016). Can Peripheral Representations Improve Clutter Metrics on Complex Scenes?
> >
> > [F] Wang et al. (2017). Central and peripheral vision for scene recognition: A neurocomputational modeling exploration.
> >
> > [G] Reddy et al. (2020). Biologically inspired mechanisms for adversarial robustness.
> >
> > [H] Jonnalagadda et al. (2021). Foveater: Foveated transformer for image classification.

---

> > > ### Author Response · Authors · 2022-08-09
> > > **Author response to reviewer XfL6 (continued)**
> > >
> > > **[Segmentation results]**
> > > In the rebuttal, we promised to evaluate our model on other downstream tasks such as object detection and semantic segmentation. However, we found that the quadratic complexity of our model makes the evaluations on time infeasible; these downstream tasks typically take input images of higher resolutions (approx. $800 \times 1000$ images) than classification, being usually performed only by the networks that process inputs within feasible memory and time complexity, e.g, linear complexity with respect to the number of input tokens as in XCiT [16]. For example, some ViT-based work in Tab.3 [16, 23, 31, 54] introduce efficient self-attention techniques such as cross-covariance attention [16], squeezed convolutional projection [23], and window-wise attentions [31, 53], thus being able to evaluate their models on detection/segmentation. Meanwhile, the relevant work of  [3, 15, 18, 25, 55] (as well as ours) with ‘quadratic complexity’ do not perform such evaluations as they typically generate out-of-memory (OOM) errors given high-resolution input images. Nevertheless, to address the reviewer’s concern, we explored a number of variants of PerViT during the rebuttal to provide the reviewers with ‘memory-efficient PerViT’ but this exploration is constrained by the limited time of the rebuttal; the ideation & exploration of memory-efficient $\Phi_p$, the implementation, pre-training on ImageNet-1K, downstream task evaluation all demand a fair amount of time and resources for the evaluations to be complete. Although we were not able to fully address the concerns, we appreciate the reviewer’s feedback that helped us discover the reason why the current implementation of PerViT is limited for other tasks of high-resolution inputs. To make our manuscript more self-contained, we will discuss this point in the limitation section (Sec.5) in our final manuscript.

---

### Official Review · Reviewer_mNPh · 2022-07-11

**Rating:** 7
**Confidence:** 3
**Soundness:** 2 fair
**Presentation:** 2 fair
**Contribution:** 3 good

**Summary:**

This paper takes inspiration from the foveated nature of human vision, in which visual scenes are processed with decreasing resolution and increased information compression as eccentricity to the center of the fovea increases.
It notes that recent vision transformers (ViTs), which use multi-headed self-attention, require a lot of data to learn useful patterns of self-attention at different layers of the feature hierarchy.
While various methods have been proposed to address this, often relying in one way or another on re-asserting locality into the processing e.g. explicitly via convolution or by introducing pyramidal structure or more structured attention, the paper here proposes adding a position-based attention mechanism into a standard data-efficient image transformer (DeIT).
In the MPA, learned position-based attention maps are combined with the (original) learned content-based attention maps to produce a  "Multi-head Peripheral Attention" (MPA) map.
Experiments on ImageNet-1k show that their Peripheral ViT (PerViT) model which uses MPA uses this additional structure in the attention to perform better at the classification task than without, to a level on par with recent pyramidal architectures, across three different approximate model sizes.
Further experiments look at the qualitative nature of the learned peripheral maps, the trade off of influence between the position-based and content-based maps, the locality of the attention at different layers, the benefit of careful initialization, and an ablation of the various model additions.

**Questions:**

In addition to the points mentioned above,
1. I think Figure 1 could be explained more clearly to inform a new reader - could you explain how does this learned attention map relate to peripheral vision?
2. How was the D_r chosen - and what is the impact of varying it?
3. L142 what is K?
4. L179 as far as I'm aware, "cylindrical" is not a standard term. What is meant by the term here? Do you mean "columnar", in the sense as described by e.g. https://arxiv.org/pdf/2102.12122.pdf ? Columnar is strictly better, because cylindrical implies a circle in some plane?
5. I'm confused why w_r is appearing in Eq 9, when it doesn't appear to be part of the peripheral projection? (based on my reading of L122-L132). Is this just describing all of the parameter settings for all experiments?
6. What does classification of visual regions add in Fig 5? It seems somewhat arbitrary, since "angle" doesn't have an obvious notion in an image taken from an unknown camera. In Supp Sec B, there is some explanation of visual fields broken down by angle, but is it fair to assume that images correspond to a particular FOV? Peripheral vision only makes sense in the context of a physical eyeball which has a focal point; how does that relate to an image from an unknown camera? An alternative might be to make a scatter plot of average attention distances vs layers, in a similar vein to Fig 3 of [36]?
7. What, if any, is the distinction between the proposed Peripheral Positional Encoding (PPE) (Table 2) and the used Convolutional Positional Encoding (CPE) (Table 1)?
8. Table 5 of the DeIT paper https://arxiv.org/pdf/2012.12877.pdf shows that using the distillation method during training yields stronger performance (e.g. DeIT-B gives 83.4 Top-1, and higher when trained for longer). Although this is not necessarily a fair comparison, I think it could be mentioned at least. The abstract and contributions claim unconditionally (L11, L69) that the PerViT yields "state of the art performance in image classification task" but I think this is not technically true and ought to be revised?
9. parts of S3.1 could be rewritten to improve readability, in particular the explanation of Peripheral Projections from L141. "By referring neighboring relative distances around the keys" does not make sense to me - could it be rewritten?
10. In Eq 8 - the PP function does not appear to be explicitly defined until the supplementary. Perhaps some of this section could be represented graphically, similar to Fig 1 in [50], to improve clarity?

**Limitations:**

Yes

**Strengths And Weaknesses:**

[Paper Strengths]
* the proposed multi-head peripheral attention approach seem to be new, technically sound, and interesting
* the experiments are well-motivated, with carefully chosen baselines and good performance
* code is available with promise of reproducible experiments (although I did not try)

[Paper Weaknesses]
Overall I think the paper presents an interesting approach backed up by some good empirical results, but there are a few reasons why I remain borderline:
* I think that the paper somewhat oversimplifies peripheral vision and oversells its involvement in the design of the model. There is only one reference to peripheral vision in the main text, and it is a bit reductive -- I encourage the authors to explore more recent literature on peripheral vision, e.g. https://visxvision.files.wordpress.com/2017/08/ruth_peripheral.pdf in order to better relate their work to peripheral vision literature. I think it is fair to be inspired by the notion of peripheral vision, but using definitive terms like "peripheral inductive bias" (L56) may be misleading. Claims like L47 "According to recent study on inner workings of vision transformers [11, 15, 36, 42, 51], their behavior is in fact closely related to how the peripheral vision functions" are not supported by the cited works (which make no mention of peripheral vision), and are arguably more speculation than scientific "fact". I believe the paper will be better served by toning down this aspect; it is ok to be inspired loosely by a notion of how peripheral vision might work, and this model might indeed model peripheral vision to a better extent than previous works, but as far as I can tell, we do not know and cannot be certain.
* Secondly, although the experiments are reasonably extensive, a few experiments are missing which I think are important for a reader to understand the value MPA. (i) One of the main motivations to impose structure into the attention (I think) is to improve training sample efficiency. Have any experiments been conducted to explore this (e.g. performance vs training set proportion on ImageNet)? (ii) My understanding is that of the three components (phi_p, C-Stem, CPE), phi_p is the main novelty of this work, while C-Stem and CPE have been adopted from prior work. I would like to see the ablation on the -S and -M models to understand the impact with/without only phi_p in those cases. (iii) Experiments on other datasets (e.g. ones which DeIT supports already like iNaturalist) would help to further validate the usefulness of the proposed method.
* Although I found the paper to have good general coverage of recent work in this area, I found the related work section to be somewhat brief, and would appreciate a more explicit explanation of why the proposed work differs from previous efforts to improve the efficacy of the self-attention mechanism for vision transformers.

EDIT: post-rebuttal, upgraded Rating from 5 to 7 (accept).

[Minor Points / typos]
* L9 "large-scale ImageNet dataset" -- "large-scale" here may mislead readers to think of the full ImageNet dataset; in reality the 1K dataset is used in the paper
* some references are not ordered numerically e.g. L30, 53, 82
* L56 - we proposes
* Fig 3 - bases on -> is based on the
* Sec 3 - please define new variables when they are introduced (e.g. D_h), for better clarity.
* Eq 3 has an unwanted comma
* L119-21 - break up the sentence for better clarity
* L123 - Eucliden
* L134 non-linearlity
* L142 the they
* please check grammar in 4.1
* L252 - double negative should be single negative?
* Please fix table ordering in 4.2 so it follows text

---

> ### Author Response · Authors · 2022-07-31
> **Author response to reviewer mNPh**
>
> We thank reviewer mNPh for professional comments and suggestions on recent peripheral vision literature and will revise the narrative accordingly to make our paper more reliable.
>
> **[Oversimplified peripheral vision]**
> Our manuscript in current form primarily focuses on developing self-attention for modeling peripheral vision but indeed oversimplifies its concept. We truly appreciate the suggested references [B, F, G, H] by reviewers RmNPh and RXfL6. With a careful review of the work and more exploration of recent literature on peripheral vision, we will revise the manuscript by updating the current (rough) narrative on peripheral vision and correcting misleading definitive terms (L47, 56) accordingly, reflecting the reviewers’ comments as much as possible. We will also extend related work in the revised manuscript, discussing recent work [B-E] on peripheral vision and how our method differs from previous efforts [F-H].
>
> **[Training sample efficiency experiments]**
> We investigate the training sample efficiency of PerViT-S by subsampling ImageNet training data by fractions of 50% and 25% and compare the results with DeiT [11] in **Tab.R4**. For each subsamples, we increase the number of epochs so the models are presented with a fixed number of images. Our model consistently surpasses the baseline [11] for all subsampled datasets, showing its robustness under limited training data. We will include the results in our final manuscript.
>
> **Table R4**. ImageNet top-1 accuracy comparisons under different subsampling ratios.
> |   Subsampling ratio   | DeiT-S top-1 | PerViT-S top-1 |
> |:----:|:------------:|:--------------:|
> | 100% |     79.9     |      82.1      |
> |  50% |     74.6     |      77.4      |
> |  25% |     61.8     |      67.5      |
>
> **[Additional ablation experiments]**
> We summarize the requested ablation on PerViT-T/S/M in **Tab.R5**; without $\Phi_p$, the top-1 accuracy consistently drops for all the three models. Comparing (b) with (c), we observe that C-Stem and CPE are less effective for large models, bringing 1.3%p and 0.1%p gains for Small and Medium respectively whereas they improve the Tiny model by 5.1%p. In contrast, the effectiveness of $\Phi_p$ is consistent across different model sizes, bringing ~1%p gains for all the three models. The effectiveness of $\Phi_p$ for larger models, we hypothesize, is due to its flexibility in modeling local/global spatial attentions (Fig.7) while C-Stem/CPE perform only locally.
>
> **Table R5**. Ablation on PerViT-T/S/M: effect of $\Phi_p$, C-Stem, and CPE.
> |                     Model                     | $\Phi_p$ | C-Stem & CPE | Tiny | Small | Medium |
> |:---------------------------------------------:|:--------:|:------------:|:----:|:-----:|:------:|
> |              (a) PerViT (ours)              |     o    |       o      | 78.8 |  82.1 |  82.9  |
> |              (b) without $\Phi_p$             |     x    |       o      | 77.3 |  81.1 |  81.9  |
> | (c) without $\Phi_p$, C-Stem, CPE (DeiT [11]) |     x    |       x      | 72.2 |  79.8 |  81.8  |
>
>
> **[Experiments on other datasets]**
> To further verify the robustness of the proposed method, we compare the PerViT-M with baseline models [11, 15] on different transfer learning task with ImageNet pre-training in **Tab.R6**. We finetune trained PerViT-M on CIFAR-10, CIFAR-100, and iNaturalist-19, following the same training recipes of DeiT [11]. Even with significantly lower complexity than  [11, 15], our method surpasses baselines by approx. 1%p on CIFAR-100 and iNaturalist19 while performing on par with [11] on CIFAR-10. Please excuse the absence of results on Flowers, Cars, and iNaturalist-18 datasets due to the tight submission deadline of the rebuttal. We will do our best to include them all in our final manuscript.
>
> **Table R6**. Transfer learning results on CIFAR-10, CIFAR-100, and iNAT-19.
> |      Model      | Size | GFLOPs | CIFAR-10 | CIFAR-100 | iNAT-19 | ImageNet-1K |
> |:---------------:|:----:|:------:|:--------:|:---------:|:-------:|:-----------:|
> |  ViT-L/16 [15]  |  307 |   117  |   97.9   |    86.4   |    -    |     76.5    |
> |   DeiT-B [11]   |  86  |   18   |   99.1   |    90.8   |   77.7  |     81.8    |
> | PerViT-M (ours) |  44  |    9   |   99.1   |    91.4   |   78.5  |     82.9    |
>
> **[Minor points / typos]**
> We appreciate the comments and will correct the typos accordingly.

---

> > ### Author Response · Authors · 2022-08-02
> > **Author response to reviewer mNPh (continued)**
> >
> > **[Clarification on our method, e.g., peripheral projections]**
> > The proposed position-based attention $\Phi_{p}$ extends the idea of previous relative positional (RP) encoding work [10, 15, 22, 37, 38, 50] which all use a single-layer linear projection, i.e., neural networks. In our work, we adopt a multi-layered design to provide torus-shaped attentions (top-right of Fig.2) and peripheral projections (PP) to break rotational symmetric properties (middle of Fig.2) to model peripheral vision. Specifically, given relative positions as inputs $\mathbf{R} \in \mathbb{R}^{HW \times HW \times D_r}$, PP with $\mathbf{W} \in \mathbb{R}^{K^2 \times D_r \times D_{\text{hid}}}$ is formally defined as follows: $PP(\mathbf{R}, \mathbf{W})_{\mathbf{q}, \mathbf{k}, :} \coloneqq \sum\_{\mathbf{m} \in \mathcal{N}(\mathbf{k})} \mathbf{R}\_{\mathbf{q}, \mathbf{m}, :} \mathbf{W}\_{\mathbf{m} - \mathbf{k}, :, :}$ where the neighbor function $\mathcal{N}$ provides a set of neighbors around given input key position $\mathbf{k}$ and is formally defined as $\mathcal{N}(\mathbf{k}) \coloneqq \left[ \mathbf{k}-[\frac{K}{2}], \dots, \mathbf{k}+[\frac{K}{2}] \right] \times \left[ \mathbf{k}-[\frac{K}{2}], \dots, \mathbf{k}+[\frac{K}{2}] \right]$. We use $K=3$ for all layers and heads of PerViT in our experiments as $K > 3$ hardly brought improvements (Supp. L201). We will rephrase this part (with a visual illustration if possible) in our final manuscript.
> >
> > **[Answers to questions]**
> >
> > 1/6: In this work, we consider each query location $\mathbf{q}$ of MPA, e.g., the position of a feature we want to transform, as a focal point, assuming each MPA in PerViT simultaneously processes $H \times W$ pixel locations with $H \times W$ different focal points given input feature size of $H \times W$. This assumption provides ring-shaped attentions if a query is located at the center of the feature map (Fig.1). While we have developed our narrative in the context of images (2D), we agree on the reviewer’s point that this assumption deviates from the reality when considering FOV of a physical eyeball (3D). We will discuss this point in our final manuscript.
> >
> > 2: For PerViT-T, we experimented with $D_r \in$ {16, 64, 256} but increasing its size hardly brought improvements, thus setting $D_r = 16$ for the Tiny model (32 and 48 for Small and Medium models resp.).
> >
> > 3/9/10: Please refer to our answers above in section **Clarification on our method, e.g., peripheral projections**.
> >
> > 4: We appreciate the suggestion and will use the term “columnar”.
> >
> > 5: Note that $w_r$ where $r \in [D_{r}]$ is a learnable parameter that weighs input Euclidean distances in $D_r$ different ways: $\mathbf{R}\_{\mathbf{q}, \mathbf{k}, :} \coloneqq \text{concat}\_{r \in D_{r}} \[w_{r} \cdot \mathbf{R}^{\text{euc}}_{\mathbf{q}, \mathbf{k}}\] \in \mathbb{R}^{D_r}$ (L125-127). Since $\mathbf{R}$ is an input to the peripheral projection in Eq.7, $w_r$ is also a part of peripheral projection, thus being utilized for the peripheral initialization in Eq.9.
> >
> >
> > 7: The proposed Peripheral Positional Encoding (PPE) refers to $\Phi_p$ in Eq.7, i.e., position information injected to self-attention matrix $\Phi_c$. The Convolutional Positional Encoding (CPE) refers to a depth-wise convolution (L192); the term CPE is originally used in the work of [7].
> >
> >
> > 8: We agree on the reviewer’s point that our claim of state of the art in the current manuscript needs to be revised. We will revise the claim in L11 and L69 accordingly as follows: “The performance improvements in image classification task over the columnar Transformer baselines, e.g., DeiT, across different model sizes demonstrate the efficacy of the proposed method.”
> >
> >
> >
> > [A] He et al. (2022). Masked Autoencoders Are Scalable Vision Learners.
> >
> > [B] Rosenholtz R. (2016). Capabilities and Limitations of Peripheral Vision.
> >
> > [C] Balas et al. (2009). A summary-statistic representation in peripheral vision explains visual crowding.
> >
> > [D] Deza et al. (2020). Emergent Properties of Foveated Perceptual Systems.
> >
> > [E] Deza et al. (2016). Can Peripheral Representations Improve Clutter Metrics on Complex Scenes?
> >
> > [F] Wang et al. (2017). Central and peripheral vision for scene recognition: A neurocomputational modeling exploration.
> >
> > [G] Reddy et al. (2020). Biologically inspired mechanisms for adversarial robustness.
> >
> > [H] Jonnalagadda et al. (2021). Foveater: Foveated transformer for image classification.

---

> > > ### Comment · Reviewer_mNPh · 2022-08-09
> > > **Upgrading recommendation**
> > >
> > > Thank you to the authors for providing an extensive rebuttal within a short time. My major concerns have been addressed, and I have read through other reviewer comments and feedback. I'm sorry that I didn't have much time to engage in discussion, but from my point of view there are no contentious topics to discuss and I believe that with the additional results and more careful framing of the contributions as mentioned in the rebuttal, the paper is a nice contribution to the NeurIPS, which others will find interesting. I am happy to update my final score to Accept.

---

> > > > ### Author Response · Authors · 2022-08-09
> > > > **Reply to reviewr mNPh**
> > > >
> > > > We again thank the reviewer for the professional, insightful suggestions and positive evaluations on our work. Based on the comments, we have updated the draft (pdf) for better presentation of peripheral vision (toning down rough narrative on peripheral vision & discussing relevant previous work) and will further spend a fair amount of time to polish the writing even after the rebuttal with exploration of more recent literature on peripheral vision.

---

### Official Review · Reviewer_NKKB · 2022-07-12

**Rating:** 5
**Confidence:** 4
**Soundness:** 3 good
**Presentation:** 3 good
**Contribution:** 3 good

**Summary:**

The paper proposes a transformer architecture which incorporates human-like peripheral vision. They modify the multi-head attention with peripheral attention, which consists of content and position attention. Content attention is the same as scaled dot product attention. Position attention is learned using a neural network which takes a fixed relative position embedding as input and is independent of the input image. The position-based attention learned by the proposed approach seems to divide the image into distinct regions similar to peripheral vision. Finally, they present quantitative and qualitative results to validate the effectiveness of their approach.

**Questions:**

* What is the formulation of neighbour function, and how does it impact the learned position-based attention?

* There should be a baseline which uses fixed position embedding varying by layer to highlight the benefits of learning position embedding using NN. Did the authors try such a baseline? Would appreciate a discussion on that in the author's response.

* Can you take the ViT implementation of [A] and apply the peripheral positional encoding on it?

**Limitations:**

Yes, the limitations are addressed adequately.

**Strengths And Weaknesses:**

Strengths:
* The paper aims to design neural networks which see in a similar way to humans. The direction and the motivations are novel.
* Extensive qualitative results help in understanding the proposed approach.

Weakness:
* The motivation behind using neural networks for learning position-based attention is not explained clearly.
* It is possibly difficult to disentengle the performance gained by the particular change (like the peripheral encodings in thie case). There is huge variation withing different implementations of ViT itself. For instance, the implementation of ViT in [A], significantly outperforms the original model. ViT implementation in [A] without autoencoder fine tuning achieves 82% top1 on imageNet, bringing over 5% improvement on the original model (76.5%). It is hence natural to question approaches which bring minor gains (less than a percent). The change can clearly be achived by simpler engineering corrections.
* A more solid approach would be to use the optimally trained ViT and then base the experiments on it.

[A] He et al. Masked Autoencoders Are Scalable Vision Learners. CVPR 2022

---

> ### Author Response · Authors · 2022-07-31
> **Author response to reviewer NKKB**
>
> We thank reviewer NKKB for constructive comments and suggestions and will revise our paper by reflecting them as much as possible.
>
> **[Clarification on peripheral projections]**
> The proposed position-based attention $\Phi_{p}$ extends the idea of previous relative positional (RP) encoding work [10, 15, 22, 37, 38, 50] which all use a single-layer linear projection, i.e., neural networks. In our work, we adopt a multi-layered design to provide torus-shaped attentions (top-right of Fig.2) and peripheral projections (PP) to break rotational symmetric properties (middle of Fig.2) to model peripheral vision. Specifically, given relative positions as inputs $\mathbf{R} \in \mathbb{R}^{HW \times HW \times D_r}$, PP with $\mathbf{W} \in \mathbb{R}^{K^2 \times D_r \times D_{\text{hid}}}$ is formally defined as follows: $PP(\mathbf{R}, \mathbf{W})_{\mathbf{q}, \mathbf{k}, :} \coloneqq \sum\_{\mathbf{m} \in \mathcal{N}(\mathbf{k})} \mathbf{R}\_{\mathbf{q}, \mathbf{m}, :} \mathbf{W}\_{\mathbf{m} - \mathbf{k}, :, :}$ where the neighbor function $\mathcal{N}$ provides a set of neighbors around given input key position $\mathbf{k}$ and is formally defined as $\mathcal{N}(\mathbf{k}) \coloneqq \left[ \mathbf{k}-[\frac{K}{2}], \dots, \mathbf{k}+[\frac{K}{2}] \right] \times \left[ \mathbf{k}-[\frac{K}{2}], \dots, \mathbf{k}+[\frac{K}{2}] \right]$. We use $K=3$ for all layers and heads of PerViT in our experiments as $K > 3$ hardly brought improvements (Supp. L201). We will clarify this in our revised manuscript.
>
> **[Experiments with different ViT implementation, e.g., MAE [A]]**
> We'd like to note that the ViT implementation in MAE are designed for large ViT models, e.g., ViT-Base/Large/Huge, not for small ViTs, e.g., ViT-Tiny/Small/Medium, that our paper adopts. According to the MAE paper [A], “ViT-L is very big and tends to overfit” (Sec.4 of [A]) and “the training is unstable with NaN frequently observed during training” (Appendix.2 of [A]) so it explores different training recipes for the large ViTs (Tab.11 of [A]) with strong regularization which brings ~5%p improvements (from 76.5 to 82.5 as seen in Sec.4 of [A]) for ViT-L. Therefore, we adopt ViT implementations of DeiT [11] not those of MAE because our paper explores small ViT models, e.g., PerViT-T/S/M; we compare the models sizes in **Tab.R1**. To see how MAE's ViT implementation affects the performance of our (small) models, we conduct experiments with PerViT-T/S/M using MAE's ViT implementations and summarize the results in **Tab.R2**; strong regularization of [A] severely damages performance for all three models. Meanwhile, we observe that PerViT-S/M less suffer compared to PerViT-T as the implementations of MAE are optimal for large ViT models.
>
> **Table R1**. Model size, e.g., the number of parameters (M), comparison.
> | PerViT-T | PerViT-S | PerViT-M | MAE-B | MAE-L | MAE-H |
> |:--------:|:--------:|:--------:|:-----:|:-----:|:-----:|
> |    7.6   |    21    |    44    |   86  |  304  |  632  |
>
> **Table R2**. ImageNet top-1 accuracy comparisons using different implementations of DeiT and MAE [A].
> |    Impl.    | Tiny | Small | Medium |
> |:-----------:|:----:|:-----:|:------:|
> | DeiT (ours) | 78.8 |  82.1 |  82.9  |
> |   MAE [A]   | 74.9 |  81.1 |  82.8  |
>
>
>
> **[Additional baselines: $\Phi_p$ with fixed parameters]**
> To highlight the benefits of learning $\Phi_p$ using NN, we conduct additional baseline experiments using PerViT-Tiny with parameters of $\Phi_p$ fixed during training under three different initialization methods (L279-283 & top sec. of Tab.4) and summarize the results in a **Tab.R3**. Fixing $\Phi_p$ damages for all three intializations which verify the efficacy of  learning diverse position-based attentions across different layers and heads. We also observe that conv and rand inits perform poorly compared to PerViT-T without $\Phi_p$, e.g., model (b) of Tab.1; we suspect that $\Phi_p$ with fixed conv & rand inits only provides local and noisy attentions respectively for all layers while the model without $\Phi_p$ has no such strong restrictions. We will include the results and discussion in our final manuscript.
>
> **Table R3**. ImageNet top-1 accuracy comparisons with fixed $\Phi_p$ under three different initializations.
> |        Model        | peri (ours) | conv | rand |
> |:--------------:|:-----------:|:----:|:----:|
> |      Ours      |     78.8    | 78.6 | 78.5 |
> | Fixed $\Phi_p$ |     77.8    | 77.1 | 75.8 |
> |   Without $\Phi_p$  |     77.3    | 77.3 | 77.3 |
>
>
> [A] He et al., Masked Autoencoders Are Scalable Vision Learners.

---

### Author Response · Authors · 2022-08-02
**General response**

We thank all the reviewers for their insightful comments and suggestions. We are glad to see that the reviewers found our work has "novel direction and motivation with extensive qualitative results (NKKB)", "interesting, technically sound, well-motivated approach with good performance (mNPh)", "concise idea with convincing and reasonable performance evaluation (XfL6)", and "complete ablation study with plenty of useful figures (5MyL)". Nevertheless, the reviewers also point out important comments stating that:

1. the method section writing can be improved for clarity,
2. the proposed method requires additional ablation study,
3. further evaluations on other datasets/tasks are missing,
4. the paper oversimplifies/oversells peripheral vision and needs to extend related work.

In the rebuttal, we clarify the method section, e.g., peripehral projection, perform additional ablation study to further verify usefulness of our approach, provide experimental results on other datasets, and promise to tone down current narrative about peripheral vision. In our final manuscript, we will do our best to reflect all the comments from the reviewers as well as additional comments given in author-reviewer discussion period.

---

### Author Response · Authors · 2022-08-09
**General response**

We thank the reviewers for reading our answers in the rebuttal and upgrading the final recommendation (mNPh and 5MyL). We appreciate reviewers NKKB, mNPh, and XfL6 for positive evaluations on our manuscript and are delighted to find that the rebuttal resolved most of the concerns for reviewer 5MyL. We have undergone a revision of the manuscript and will further spend a fair amount of time to polish the writing (with additional experiments if possible) after the rebuttal, e.g., exploration on recent work of peripheral vision, toning down rough narrative, and experiments on datasets like iNaturalist-18. In the revised manuscript, the revised/added texts & experiments are colored in blue. Due to the 9-page limit for the rebuttal revision (https://nips.cc/Conferences/2022/PaperInformation/NeurIPS-FAQ), we leave out some original texts (colored in red) in the current revised manuscript; we will include the texts in our final manuscript (10-page limit) if accepted. Even after the rebuttal, we will be highly appreciated for further suggestion and feedback to make our manuscript more reliable and stronger.

We replace the original manuscript with revised one (pdf).
Our orignal main/supplementary manuscripts and submitted code can be found in the supplementary material (zip).

---

### Meta-Review · Area_Chair_QWJU · 2022-08-24

**Recommendation:** Accept
**Confidence:** Certain

**Metareview:**

The paper proposes a transformer architecture that models human-like peripheral vision. Experiment results show it achieves good performance. All the reviewers consider the paper above the bar. They like the novelty and the strong empirical performance. The AC finds no reason to object.

**Award:**

No

---

### Decision · Program_Chairs · 2022-09-14

Accept